**Predicting Ice Supersaturation for Contrail Avoidance: Ensemble** Forecasting using ICON with Two-Moment Ice Microphysics

Maleen Hanst<sup>1</sup>, Carmen G. Köhler<sup>1</sup>, Axel Seifert<sup>1</sup>, and Linda Schlemmer<sup>1</sup>

<sup>1</sup>Deutscher Wetterdienst, Frankfurter Straße 135, 63067 Offenbach am Main, Germany

**Correspondence:** Maleen Hanst (maleen.hanst@dwd.de)

Abstract. Persistent contrails and contrail-induced cirrus clouds are considered the most significant non-CO<sub>2</sub> contributors to aviation's climate impact. These clouds primarily form in ice-supersaturated regions (ISSRs), defined by relative humidity over ice (RH<sub>ice</sub>) exceeding 100 %. Reliable prediction of RH<sub>ice</sub> in the upper troposphere and lower stratosphere allows mitigating their formation by re-routing flights. We implemented a two-moment cloud ice microphysics parameterization within a tenmember Ensemble Prediction System (EPS) using the global ICON (ICOsahedral Nonhydrostatic) model. RHice predictions were evaluated against radiosonde and aircraft observations from the Northern Hemisphere during 2024-2025. Treating ISSR prediction (RH<sub>ice</sub> > 100 %) as a binary classification problem, we find that the probability of detection (POD) of ISSRs increases to 0.6 for the two-moment scheme (ICON 2-Mom), compared to 0.4 for the operational ICON with a one-moment ice microphysics scheme, while maintaining a low false positive rate (FPR < 0.1). Further evaluation of the ICON 2-Mom EPS using Receiver Operating Characteristic (ROC) analysis shows a POD of 0.8 for a decision model that requires at least three ensemble members to predict ISSR, with an FPR of 0.13. Additionally, we incorporate ensemble spread information to develop a meta-model that further reduces the FPR. Since June 2024, more than 100 flights have been rerouted based on ICON 2-Mom EPS predictions in a contrail avoidance trial, demonstrating the practical value of improved ISSR forecasts for climate-conscious aviation. This study highlights the significant potential of ensemble-based modeling for predicting ISSRs and RH<sub>ice</sub>, supporting environmentally optimized flight planning and advancing applications in weather and climate science.

Copyright statement. TEXT

#### 1 Introduction

The impact of aviation on climate change is a growing concern, especially as the number of aircraft increases (Yamashita et al., 2016; Grewe et al., 2021). Air traffic is estimated to contribute to global warming by approximately 3.5 % (Lee et al., 2023) – with an uncertainty range of 2–14 % (Lee, 2018) – caused by CO<sub>2</sub> and non-CO<sub>2</sub> effects.

While the uncertainty range for the climate impact of CO<sub>2</sub> emissions is relatively small, there is significant variability associated with non-CO<sub>2</sub> effects arising from emissions such as  $NO_x$ ,  $H_2O$ , and, notably, the formation of persistent contrails and contrail-induced cirrus clouds (Matthes et al., 2017; Klöwer et al., 2021; Lührs et al., 2021; Lee et al., 2023). These aircraftinduced clouds present a complex challenge for climate assessment (Teoh et al., 2024). While Kärcher (2018) estimates that they account for more than half of aviation's total radiative forcing, Bickel et al. (2025) contend that their net warming effect might be less than that of CO<sub>2</sub>, primarily because it may be partially offset by a decrease in natural cirrus cloud coverage (Bickel et al., 2020).

Given the variety of findings and the potential trade-off between CO<sub>2</sub> and non-CO<sub>2</sub> impacts, effective strategies to mitigate the climate impact of aviation must address both types of effects. One such strategy that has gained increasing attention in recent years is climate-optimized flight routing, which aims to reduce aviation-induced warming by accounting for a comprehensive range of atmospheric impacts (Schumann et al., 2011; Grewe et al., 2017a, b; Matthes et al., 2017; Simorgh et al., 2022). This approach is built upon climate response models such as the Contrail Cirrus Prediction (CoCiP) model (Schumann, 2012), its Python adaptation PyContrails (Shapiro et al., 2023), or algorithmic Climate Change Functions (aCCF) (Dietmüller et al., 2022; Matthes et al., 2023), which provide the necessary computational framework.

Climate response models rely on four-dimensional meteorological fields – typically derived from numerical weather prediction (NWP) models – in which relative humidity over ice (RH<sub>ice</sub>) is a key parameter for evaluating contrail formation according to the Schmidt-Appleman criterion (Schmidt, 1941; Appleman, 1953; Schumann, 1996). To provide climate response models with physically consistent and representative atmospheric inputs, it is crucial that NWP models accurately capture RH<sub>ice</sub>, especially under ice-supersaturated conditions (RH<sub>ice</sub> > 100%), which are essential for persistent contrail development.

35

40

Beyond contrail modeling, ice-supersaturated regions play a critical role in the development and persistence of cirrus clouds, which are key regulators of the water vapor budget in the upper troposphere and lower stratosphere (Kärcher et al., 2023). Improving the representation of supersaturation is therefore vital not only for contrail modeling but also for capturing the broader impacts of cirrus cloud dynamics on atmospheric moisture and radiative balance (Dekoutsidis et al., 2023; Borella et al., 2025). Yet, despite its relevance for climate-relevant processes, RH<sub>ice</sub> remains one of the most uncertain variables in NWP models (Kunz et al., 2014; Dyroff et al., 2015; Krüger et al., 2022).

 $RH_{ice}$  prediction is particularly challenging due to limited upper tropospheric humidity data, large humidity variability, and incomplete understanding of ice nucleation and cirrus cloud formation. Improving cloud cover schemes and parameterizations of ice microphysics are therefore an active area of research (Kärcher et al., 2022; Seifert et al., 2022; Spichtinger et al., 2023; Achatz et al., 2024; Grundner et al., 2024; Lüttmer et al., 2024). Additionally, predicting ice supersaturation is complicated by resolution limits: NWP models represent mean atmospheric values and often miss localized ice supersaturated regions (ISSRs), especially those linked to unresolved mesoscale gravity waves (Wilhelm et al., 2018).

One way to circumvent these limitations is to develop machine learning methods to derive  $RH_{ice}$  forecast corrections. The resulting correction model receives variables such as temperature,  $RH_{ice}$ , and others, and returns adjusted values of  $RH_{ice}$ . Wang et al. (2025) focused their research on reanalysis data, deriving their post-processing model inputs from ERA5 (ECMWF Reanalysis v5) data, and trained their model using humidity measurements from the In-service Aircraft for a Global Observing System (IAGOS), showing  $RH_{ice}$  mean absolute error improvements when validated against test data. Previous studies have also examined corrections to ERA5 reanalysis  $RH_{ice}$ , particularly in the context of estimating the climate effects of aviation contrails (e.g., Teoh et al., 2022).

The use of high-resolution NWP models is another approach to dealing with uncertainties in predicting  $RH_{ice}$ . In a recent study by Thompson et al. (2024), several NWP models were validated with respect to  $RH_{ice}$  using radiosonde and IAGOS data, in the context of contrail avoidance flight routing.  $RH_{ice}$  predictions from IFS (Integrated Forecasting System), GFS (Global Forecast System), and S-WRF (a Weather Research and Forecasting model configuration by SATAVIA) were evaluated using standard classification metrics, including the  $F_1$  score and the Matthews Correlation Coefficient, which reflect the models' ability to correctly identify ice-supersaturated conditions. Moderate scores were found, indicating room for improvement in ISSR prediction skill.

The study highlights that a correct identification of non-ISSR is also crucial, as false negatives (thus, incorrect ISSR predictions) could potentially lead to unnecessary re-routing. For the S-WRF model, they find a true positive rate for the non-ISSR condition of 90.7 % and hence a false positive rate of ISSR of 9.3 %. The low false positive rate of ISSR suggests that there may be only few worst-case scenarios where aircraft are diverted to an incorrectly predicted non-ISSR due to an incorrectly predicted ISSR, resulting in both additional  $CO_2$  emissions and possible contrail formation.

These studies demonstrate the potential of machine learning models and state-of-the-art NWP systems to improve  $RH_{ice}$  prediction, but they also reveal persistent limitations. In particular, the reliance on simplified cloud cover or microphysics schemes and deterministic forecasts restricts the ability of current models to capture the full variability and uncertainty associated with ice supersaturation.

A key challenge in realistically representing RH<sub>ice</sub> in NWP models lies in the treatment of subgrid-scale humidity variability and cloud formation processes. The IFS model addresses this through the Tompkins cloud cover scheme (Tiedtke, 1993; Tompkins et al., 2007; ECMWF, 2024), which employs a prognostic probability distribution function (PDF) of total water content to estimate cloud fraction. This statistical approach allows for a probabilistic representation of cloud cover and ice supersaturation but does not explicitly resolve the underlying ice microphysical processes.

75

80

90

In contrast, the ICON (ICOsahedral Nonhydrostatic) NWP model (Zängl et al., 2014) uses a physically based microphysics scheme. Within this approach, a key factor in a realistic representation of RH<sub>ice</sub> is the scheme's ability to simulate the phase relaxation time – the timescale over which water vapor transitions to ice. In the operational one-moment cloud ice microphysics parameterization, the specific ice mass is treated as a prognostic variable, whereas ice particle number density is estimated from temperature. This approach tends to overestimate particle numbers at low temperatures, resulting in unrealistically short phase relaxation times and limiting the ability of the model to represent ice supersaturated conditions. To address these limitations, a two-moment ice microphysics scheme treats the ice particle number density as an additional prognostic variable (Köhler and Seifert, 2015). This allows ICON to better capture the phase relaxation time, and thereby the degree of ice supersaturation and the persistence of ice supersaturated regions.

Complementary to the model physics, ensemble forecasting is a powerful tool for capturing atmospheric variability and model uncertainty. This is particularly valuable for phenomena like ISSR, which are rare, spatially heterogeneous, and sensitive to small-scale processes. While ensemble means are commonly used to produce stable deterministic forecasts, they may obscure signals critical for ISSR detection. Instead, ensemble spread and extremes, such as the highest RH<sub>ice</sub> values among members, may reveal localized supersaturation events and offer a probabilistic measure of forecast confidence.

In our study, we combine both approaches: We implement and evaluate a two-moment cloud ice microphysics scheme in the global ICON. Further, we explore its impact within a ten-member ensemble prediction system (EPS), assessing how the ensemble can enhance ISSR identification beyond mean-state representation.

As part of this study, the ten-member ICON ensemble with the new two-moment ice microphysics scheme has been established as a dedicated forecasting system at the German Meteorological Service (DWD). It provides continuous meteorological data to support research on contrail avoidance flights. This setup was developed within the D-KULT project (Demonstrator for Climate and Environmentally Friendly Air Transport), which aims to demonstrate the feasibility of climate-optimized flight trajectories with a focus on reducing contrail formation in European airspace. The project seeks to optimize flight paths using climate response models that account for both CO<sub>2</sub> and non-CO<sub>2</sub> effects, while balancing emissions, noise, operating costs, and operational constraints such as airspace regulations and airport capacity. One key component is the integration of forecasts from the ICON ensemble to identify potential persistent contrail regions for contrail avoidance planning. In real-world trials, more than 100 flights have already been rerouted using information from these forecasts, demonstrating the practical applicability of climate-aware flight operations.

The structure of this work is as follows: In Section 2, we introduce the new two-moment cloud ice microphysics parameterization, the ensemble generation, and the details of the model setup. Section 3 provides an overview of the in-situ observational data used for verification. The verification methodology is described in Section 4, and the results are presented in Section 5, where we evaluate the deterministic ICON model with the new two-moment scheme, particularly in comparison to the operational ICON model with the one-moment scheme. Building on the deterministic model verification, we then assess the ten-member ensemble setup. We conclude with a discussion in Section 6 and final remarks in Section 8.

#### 2 Model

105

110

The ICON model, developed by the ICON partnership including Deutsches Klimarechenzentrum, Max Planck Institute for Meteorology, Karlsruhe Institute of Technology, the Center for Climate Systems Modeling, and the German Meteorological Service (DWD), is used operationally by DWD. In its global operational configuration, cloud ice microphysics is represented by a one-moment scheme, where specific ice mass is prognostic and ice particle density is diagnosed from temperature (see Appendix A for details).

### 2.1 Two-Moment Cloud Ice Microphysics Parameterization in ICON

The two-moment cloud ice scheme in ICON is an extension of the operational one-moment cloud ice scheme. It adds a prognostic equation for cloud ice number density and includes explicit ice nucleation processes. Examples of similar hybrid schemes include those by Reisner et al. (1998) and Thompson et al. (2004), though these originally used purely temperature-dependent ice initiation. Köhler and Seifert (2015, hereafter KS15) present a two-moment scheme that accounts for deposition nucleation based on ice supersaturation, and includes homogeneous freezing of sulfate aerosol droplets at low temperatures. The version of the two-moment scheme used in this study is a simplified and updated version of KS15. The two-mode representation in

KS15 is omitted for computational efficiency, as are the timestep refinements for homogeneous nucleation. In a two-moment scheme, sources and sinks of ice particles must be explicitly parameterized. The three primary sources of ice particles are detrainment of ice from deep convective clouds, homogeneous nucleation, and heterogeneous nucleation.

#### **Deep Moist Convection**

30 ICON parameterizes moist convection using a bulk mass flux convection scheme (Tiedtke, 1989; Bechtold et al., 2008). For cloud ice detrainment from convection, a mean particle diameter of  $D_{i,\text{conv}} = 200 \ \mu\text{m}$  is assumed, corresponding to a mean mass of  $m_{i,\text{conv}} = 10^{-9} \text{ kg}$ . A smaller mean mass would increase the number of detrained ice particles in the upper troposphere, leading to shorter phase relaxation times in convective anvils and reduced ice supersaturation. The assumed size also affects the effective radius of anvil clouds explicitly represented in the model.

# 135 Homogeneous Ice Nucleation

For homogeneous ice nucleation, the parameterization by Kärcher et al. (2006) is used. It accounts for the presence of preexisting ice particles and is applied using grid-scale vertical velocity and ice supersaturation. However, it neglects subgrid-scale variability, which may lead to an underestimation of nucleation events. The impact on cloud ice number concentration is less straightforward. While nucleation events in nature occur on much smaller spatial scales, the model assumes that nucleated particles are evenly distributed across the grid box once the event is triggered.

### **Heterogeneous Ice Nucleation**

Heterogeneous nucleation is represented using the INAS (Ice Nucleating Active Sites) approach of Ullrich et al. (2017), which includes parameterizations for deposition and immersion freezing on mineral dust and soot. Since prognostic aerosol fields are not available in ICON, but only in ICON-ART, a constant dust number concentration of  $N_{\rm dust} = 1 \times 10^3 \ {\rm m}^{-3}$  is assumed in the upper troposphere above  $p_0 = 200 \ {\rm hPa}$ . Below that pressure height the profile increases following

$$N_{\text{dust}}(p) = N_{\text{dust},0} \max \left\{ \min \left[ \exp \left( \gamma_{\text{dust}} \frac{p}{p_0} \right), 200 \right], 1 \right\}$$
 (1)

with  $\gamma_{\text{dust}} = 1 \times 10^{-3}$ . The dust surface area  $\bar{S}_{\text{dust}}$  is calculated based on a lognormal particle size distribution with a mean diameter of 1  $\mu$ m and a standard deviation of 2.5. The number of nucleated ice particles is then diagnosed as:

$$N_i^* = N_{\text{dust}} \left\{ 1 - \exp\left[ -\bar{S}_{\text{dust}} n_S(T, S_i) \right] \right\}$$
 (2)

Here,  $n_S$  is the INAS density in m<sup>-2</sup>, parameterized according to Eq. (7) for deposition and Eq. (5) for immersion freezing in Ullrich et al. (2017).

In numerical models, newly formed ice particles are typically diagnosed each timestep using  $\Delta N_i = N_i^* - N_i^{\text{pre}}$ , where  $N_i^{\text{pre}}$  is the number of pre-existing ice particles. However, this can overestimate heterogeneous nucleation since  $N_i^{\text{pre}}$  is reduced by sedimentation or aggregation, while  $N_{\text{dust}}$  remains constant. This effectively creates an unlimited reservoir of ice-nucleating

Figure 1. Relative humidity over ice ( $RH_{ice}$ ) of the operational ICON with one-moment ice microphysics scheme (top row) and of ICON with two-moment ice microphysics scheme (bottom row): (a) Global forecast-only data of  $RH_{ice}$  near the tropopause ( $\sim$ 10.2 km); (b) normalized histograms of  $RH_{ice}$  of Vaisala RS41 radiosonde data and ICON; (c) 2D histograms of  $RH_{ice}$  of spatio-temporally matched points between Vaisala RS41 radiosonde data and ICON forecasts with a lead time of 12 hours; heights between 8500–12500 gpm, corresponding to most common commercial flight altitudes.

particles. To avoid this artifact, a budget variable is introduced as described in KS15. A relaxation timescale of two hours is applied to simulate the recovery of nucleating particle availability due to atmospheric mixing.

#### 2.2 Ensemble Generation



The ensemble generation is based on the Local Ensemble Transform Kalman Filter (LETKF) method (Ott et al., 2004; Hunt et al., 2007), which perturbs the initial conditions of all members simultaneously in a member-dimensional space. The initial state of each ensemble member is computed by combining its background state – a short-range forecast – with a weighted correction derived from the differences between observations and model background. These weights are computed via a gain matrix that incorporates both observation error and background error covariances, ensuring that each member assimilates observation information in a distinct but dynamically consistent way.

In addition to initial condition perturbations, the system includes stochastic perturbations of selected physical parameterizations which are known to be sensitive. Thereby, different components of the system are perturbed, including gravity waves, convection, microphysics, the cloud scheme, turbulence and land surface. For example for convection, well-known parameters such as the entrainment rate or the excess of moisture or temperature used in the ascent of a test parcel are targeted. For the

global ensemble system, these physical parameters are randomly perturbed for each ensemble member with time-dependent perturbations varying sinusoidally within their range. The randomisation is accomplished by a phase shift of the sinusoidal wave depending on the ensemble member ID (for more details see Chapter 13.2 in Reinert et al., 2025). This approach introduces variability among ensemble members while preserving the consistency of individual forecast trajectories. The combined perturbation strategy ensures a realistic representation of forecast uncertainty, which is crucial for assessing the sensitivity of contrail formation potential to meteorological variability.

As a third source of uncertainty, the sea-surface temperatures over oceans are perturbed in the initial conditions.

### 175 **2.3 Model Setup**



The dedicated ICON forecasting system which is implemented and evaluated in this study is based on ICON version 2.6.6. The system runs on the ICON R3B06 grid, which has a horizontal spacing of about 26 km and a vertical spacing of about 200–300 m at the most common commercial flight altitudes of 8.5–12.5 geopotential kilometers. It starts from the operational analysis, which is based on the one-moment ice microphysics scheme, so that we require a spin-up time of at least 6 hours in our evaluations below to build up ice supersaturation. The model is run four times a day, initialized at 00, 06, 12, and 18 UTC with a forecast lead time of 60 hours, producing hourly forecasts. The system consists of ten ensemble members, whose generation is based on the first ten members of the operational ensemble prediction system. This is a reasonable approach as discussed in Appendix B.

The model outlined forms the basis for the evaluations performed in this study and will be referred to as ICON 2-Mom EPS in the remainder of this study. Since the dedicated ICON forecasting system does not consist of an additional deterministic model run, we use individual members of the ensemble as approximates to a deterministic model setup for our evaluation, denoted by ICON 2-Mom in the following. Similarly, the operational ICON with the one-moment ice microphysics scheme is denoted by ICON 1-Mom.

Figure 1(a) illustrates the difference in model behavior between ICON 1-Mom and ICON 2-Mom, showing global patterns of relative humidity over ice for both schemes. While overall cloud structures remain comparable, the two-moment cloud ice scheme produces a markedly higher degree of ice supersaturation. The realism of this behavior is examined in the remainder of this study through comparison with observational data.

### 3 Observation Data

This study emphasizes in situ measurements for verification, with the primary analysis based on radiosonde data. Additionally, data from the In-Service Aircraft for a Global Observing System (IAGOS; see https://www.iagos.org/) were considered.

### 3.1 Vaisala RS41 Radiosonde Data

We restricted our radiosonde verification to Vaisala Radiosonde RS41 data, as this type of radiosonde is best scored for humidity measurements in the UTLS (Dirksen et al., 2022; Borg et al., 2023; WMO, 2024). The temperature is measured with an

**Figure 2.** Radiosonde (left) and IAGOS (right) observation data. (a) Locations of 105 stations equipped with Vaisala RS41 radiosondes in the Northern Hemisphere. (b) Example height profiles of temperature and RH<sub>ice</sub> from Vaisala RS41 (TEMP) observations and ICON 2-Mom EPS forecasts with a lead time of 12 h. (c) Rank histogram: For each spatio-temporal point (comprising ICON 2-Mom EPS values and the corresponding radiosonde measurement), the observed value is ranked among the ten ensemble members, and the resulting ranks are displayed in a histogram. The rank histogram includes only samples where the observed RH<sub>ice</sub> exceeds 50 %. (d) IAGOS flight routes of 188 flights from December 2024, limited to the Northern Hemisphere. (e) Spatio-temporal comparison of flight data and ICON 2-Mom EPS: Time series of temperature and RH<sub>ice</sub> from one example flight. (f) Rank histogram for IAGOS flight data, analogous to (c).

accuracy of  $\pm 0.2$  °C and the humidity with an accuracy of  $\pm 3$  % RH. For more details on techniques and precision compare Vaisala (2013). We limited our verification to the Northern Hemisphere, where 105 radiosonde stations frequently yield Vaisala RS41 data. In Figure 2(a), the radiosonde locations are shown. Radiosonde observations are typically conducted twice daily, with balloon ascents around 0 UTC and 12 UTC. The resulting data are stored in standardized binary files known as Binary Universal Form for the Representation of meteorological data (BUFR), a format developed by the World Meteorological Organization (WMO) to encode and transmit various types of weather observations. These files contain TEMP reports, which include a structured set of atmospheric measurements such as temperature, pressure, humidity, and wind speed and direction at multiple vertical levels. TEMP BUFR files serve as the standardized source of radiosonde data used in this study.




The Vaisala RS41 radiosondes used in this study record vertical profiles with a height resolution of approximately 1 geopotential meter (gpm) and a measurement accuracy of  $\pm 10$  gpm. Within the standardized TEMP BUFR files, dew point temperature is provided, from which RH<sub>ice</sub> is derived following the method outlined in Appendix C. Figure 2(b) illustrates example vertical profiles of temperature and RH<sub>ice</sub> obtained from radiosonde measurements, shown alongside the corresponding ICON 2-Mom EPS data.

Figure 3. Overview of categorical verification methods used in this study. (a) Confusion Matrix: Provides a structured summary of how model predictions align with actual observations in a binary classification setting. Each prediction is categorized as a true positive (TP), false positive (FP), false negative (FN), or true negative (TN), depending on its agreement with the observed outcome. This matrix forms the foundation for computing categorical performance metrics such as listed in (b). (b) Categorical metrics: Frequency bias index (FBI), probability of detection (POD), false positive rate (FPR), precision, and the Matthews correlation coefficient (MCC) offer distinct insights into model behavior as described in Section 4.2. (c) Categorical evaluation of the ensemble prediction system: (i) the discrimination diagram shows two distributions of forecast probabilities; one for the case where the event was observed in the measurements, and one where it was not observed, highlighting the model's ability to separate events by probability; (ii) the Receiver Operating Characteristic (ROC) curve illustrates the trade-off between the POD and FPR across different classification models based on the ensemble's probabilistic event forecast.

#### 3.2 IAGOS Near-Real-Time Data



In addition to radiosonde data, we use in-situ aircraft data for our verification. The In-service Aircraft for a Global Observing System (IAGOS) is a European research infrastructure that uses commercial aircraft to collect atmospheric data. IAGOS-CORE contains several measurement instruments, e.g., for ozone, carbon monoxide, humidity, and cloud particles, and optionally for nitrogen oxides, greenhouse gases, and more (https://iagos.aeris-data.fr/instrumentation/). The time resolution of the temperature measurements is 4 s with an accuracy of  $\pm 0.5$  K, while the time resolution of the humidity measurements ranges from 1 s at 300 K to 120 s at 200 K, with an accuracy of  $\pm 6$  % (for more details, see www.iagos.org/iagos-core-instruments/h2o/). There are several levels of data processing, from which we have used near-real-time (NRT) data, where humidity measurements are subject to automated quality control, usually within 72 hours (https://iagos.aeris-data.fr/levels/). Only data with validity flag "good" were used (https://iagos.aeris-data.fr/data-quality/) for 625 flights between August 2024 and January 2025. Fig. 2(d) shows the flight routes for December 2024. For an example highlighted flight route, the temperature and RH<sub>ice</sub> time series are shown together with the corresponding ICON 2-Mom EPS data (Fig. 2(e)). Similar to the radiosonde verification, the analysis is confined to the Northern Hemisphere.

#### 225 4 Verification Methods

#### 4.1 Spatio-Temporal Matching of Model and Observation Data

The ICON grid used in our model setup has a horizontal resolution of approximately 26 km and a vertical resolution of 200–300 m within the altitude range of 8500–12500 gpm. ICON simulations with hourly forecasts up to a lead time of 60 hours were started in 6 hour intervals.

### 230 Radiosonde Data

Radiosonde data from a given station are mostly horizontally fixed and provide dense vertical coverage. To generate matched ICON–radiosonde data pairs, the ICON grid cell center closest to each radiosonde station was first identified. Subsequently, radiosonde observations were linearly interpolated to the ICON levels, as the model provides mean values across vertical layers with considerably lower resolution than the radiosonde data. No horizontal interpolation was applied. However, the impact is expected to be minimal, as typical horizontal scales of ISSRs are on the order of 140 km (Spichtinger and Leschner, 2016).

For temporal matching, the start time of the accent was used as a reference, and we select the corresponding ICON simulation whose initial time is closest to the observation time minus the required lead time. Since the simulation provides hourly forecasts, this approach ensures temporal matching to the nearest hour. The exact lead time is explicitly stated in all evaluations and never below the required spin-up time of 6 hours.

Over the 14-month verification period, this approach yielded approximately 820 000 spatio-temporal matching points from more than 63 000 radiosonde profiles. Figure 2(b) shows example radiosonde profiles of temperature and RH<sub>ice</sub> from one station, compared with ICON ensemble values.

### **IAGOS Data**

IAGOS data represent aircraft-based observations and thus capture horizontal trajectories spanning several hours. Matched ICON-IAGOS data pairs were generated by identifying all ICON grid cell centers that were nearest to at least one point along each flight path. Each selected ICON cell was then paired with its closest flight data point, and the model data were vertically interpolated to match the altitude of that observation. For temporal matching, the minimum lead time was fixed at 6 hours to account for the required ICON spin-up. Since flights span several hours, different ICON simulations were used, each selected based on the initial time closest to the observation time minus the 6-hour lead time. As ICON simulations are initialized in 6-hour intervals, this approach may result in a maximum temporal mismatch of ± 3 hours.

Over the four-month verification period, this procedure yielded approximately  $200\,000$  spatio-temporal matching points from 625 flights. Figure 2(e) shows a sample time series of temperature and  $RH_{ice}$  from an intercontinental flight, together with the corresponding ICON ensemble values.

#### 4.2 Categorical Metrics

Instead of analyzing the full continuous range of RH<sub>ice</sub>, the values can be partitioned based on a specified threshold. This results in a binary classification, distinguishing between two complementary events:

 $RH_{ice} > threshold \quad or \quad RH_{ice} \leq threshold.$ 

In addition to the duration of ISSRs, pronounced ice supersaturation has been associated with the persistence of contrails (Teoh et al., 2022). While this link is relatively weak, relative humidity remains the dominant factor in contrail-cirrus evolution, governing both the total ice mass and total extinction (Unterstrasser and Gierens, 2010). Given its relevance, this study focuses on ice supersaturation events ( $RH_{ice} > 100\%$ ) and on cases of pronounced supersaturation ( $RH_{ice} \gg 100\%$ ). These are treated as the positive events in our categorical verification framework, which we particularly aim to distinguish from their complementary cases. The spatio-temporal matching points between model output and observational data can then be categorized with respect to the positive event. Positive predictions and negative predictions are classified with respect to the observed condition as true positives (TP), false negatives (FN), false positives (FP), and true negatives (TN). The results are indicated in a confusion matrix (see Figure 3(a)), which serves as the basis for computing categorical metrics (see Figure 3(b)). Below we provide all metrics which we later use to evaluate the performance of ICON 2-Mom (EPS).

The frequency bias index (FBI) is defined as the ratio of the forecast frequency of an event to its observed frequency

$$FBI = \frac{TP + FP}{TP + FN}.$$



It indicates whether the forecast system tends to overforecast (FBI > 1) or underforecast (FBI < 1) a given event.

The Probability of Detection (POD, also known as sensitivity) evaluates the forecast system's ability to correctly identify observed events. POD is defined as the ratio of correctly predicted events to the total number of observed events, given by

$$POD = \frac{TP}{TP + FN}.$$

The false positive rate (FPR, also defined as 1-specificity) quantifies the proportion of non-events that were incorrectly forecast as events. It is defined as

$$FPR = \frac{FP}{FP + TN}.$$

POD and FPR are both computed relative to the ground truth: the former with respect to the number of observed events, and the latter with respect to the number of observed non-events. To complement these metrics, precision provides a forecast-centric perspective, highlighting the trustworthiness of predicted events and if defined by

$$Prec = \frac{TP}{TP + FP}$$
.

The Matthews correlation coefficient (MCC) is a composite measure that accounts for all four components of the confusion matrix simultaneously. MCC is particularly well-suited for datasets with class imbalance (in our case we have about 13% ISSR

events), as it reflects the quality of binary classifications regardless of event prevalence. It is defined as

$$MCC = \frac{TP \cdot TN - FP \cdot FN}{\sqrt{(TP + FP)(TP + FN)(TN + FP)(TN + FN)}}.$$

The MCC ranges from -1 to +1, where +1 indicates perfect discrimination between events and non-events, 0 reflects random predictive skill, and -1 represents complete misclassification.

### 4.3 Categorical Verification of Probabilistic Model

Ensemble forecasts provide a distribution of values for any forecast quantity of interest. For binary events such as ice supersaturation, the forecast probability is defined as the fraction of ensemble members predicting the event.

# 290 Discrimination diagram



The discrimination diagram visualizes two conditional distributions of the forecast probabilities: one conditioned on the event being observed in the measurement data, and the other conditioned on the event not being observed.

To assess the discriminative capability of the EPS, we employ the discrimination diagram, which visualizes two conditional distributions of the forecast probabilities: one conditioned on the event being observed in the measurement data, and the other conditioned on the event not being observed.

These distributions are represented as normalized histograms of the EPS forecast probabilities. A clear separation between the two distributions indicates strong discriminability, reflecting the ensemble's ability to assign higher probabilities to observed events and lower probabilities to observed non-events. This method provides a threshold-independent diagnostic of classification performance in a probabilistic forecasting framework. An example sketch of a discrimination diagram is provided in Figure 3(c)(i).

# Receiver Operating Characteristic (ROC) Curve

The ROC curve is a powerful threshold-dependent verification tool to evaluate the performance of a binary classification model. Such a model typically predicts not just a binary label directly, but rather a scalar score (in our context, this score is the predicted event probability). A score is turned into an event prediction if it is above a certain threshold. The threshold itself becomes part of the model; by varying the threshold, we effectively obtain a multitude of models, each with its own POD and FPR. The ROC shows the POD versus the FPR for all of these models at once. The top-left corner corresponds to a perfect classification model. An example sketch of a ROC curve is provided in Figure 3(c)(ii).

#### 5 Verification Results

We evaluate the RH<sub>ice</sub> predictions of ICON equipped with the new two-moment ice microphysics scheme in two steps. First, we verify the deterministic model, ICON 2-Mom, which includes a comparison with ICON 1-Mom. Second, we evaluate the ensemble prediction system, ICON 2-Mom EPS.

Radiosonde data were used unless the use of IAGOS data is indicated. Only data within the 8.5–12.5 km geopotential height range were included to match commercial flight altitudes.

### 5.1 Verification of Deterministic Model ICON 2-Mom

# 315 5.1.1 Relative Frequency Distribution of RH<sub>ice</sub>

Figure 1(b) displays the relative frequency distributions of the observed  $RH_{ice}$  compared to the corresponding model-based distributions from the operational ICON 1-Mom (top) and the new ICON 2-Mom (bottom) configurations. Pronounced differences emerge in the tail of the density distribution, which reflects ice supersaturation. ICON 1-Mom exhibits a sharp peak near 100 %, followed by a rapid decline, with maximum  $RH_{ice}$  values reaching only  $\approx$ 103 %. In contrast, ICON 2-Mom more accurately captures the tail structure, slightly overshooting at low supersaturation but successfully reproducing the upper range, including  $RH_{ice}$  values up to 135 %. A few higher values were excluded from the plot due to axis truncation, ensuring comparability without distortion from rare outliers.

### 5.1.2 Continuous Spatio-Temporal Comparison

We examined the 2D histograms of RH<sub>ice</sub> of spatio-temporally matched points between Vaisala RS41 radiosonde data and ICON forecasts (Fig. 1(c)). While ICON 2-Mom reproduces the observed supersaturation range reasonably well – unlike ICON 1-Mom – noticeable scatter remains around the one-to-one line. However, perfect agreement between modeled and observed RH<sub>ice</sub> values is not strictly required in our context. Crucially, the model must reliably distinguish between ISSR events and non-events, as both have significant operational implications for flight planning and routing. To assess this capability, we proceed below with a verification based on categorical performance metrics.

#### 330 5.1.3 Categorical Verification

In the remainder of this study, we consider events of the type

 $RH_{ice} > threshold,$ 


with threshold  $\in \{100\%, 105\%, 110\%, 120\%\}$ .

Figure 4(a) compares the FBI between ICON 1-Mom and ICON 2-Mom for these events. For the ISSR event (blue curves), the FBI is slightly above 1 for ICON 2-Mom, indicating a modest overprediction, whereas ICON 1-Mom exhibits lower values around 0.75, reflecting underprediction. In both configurations, the FBI remains relatively constant across the examined altitude range. At higher RH<sub>ice</sub> thresholds, the FBI for ICON 2-Mom is slightly below 1 at lower heights but rises to a maximum of approximately 1.5 near 12 km for the event RH<sub>ice</sub> > 120 %. In contrast, ICON 1-Mom yields an FBI of zero across the entire height range, indicating a failure to detect high supersaturation events. These results demonstrate that the two-moment scheme not only predicts ice supersaturation more frequently than the one-moment scheme – which consistently underestimates event occurrence – but also tends to slightly overestimate observed event frequency.

Figure 4. Categorical verification of ICON 1-Mom and ICON 2-Mom against Vaisala RS41 radiosonde measurements. The analysis covers data from the Northern Hemisphere within the most frequently flown altitude range of 8.5–12.5 km geopotential height, over a verification period of 11.5 months (June 15, 2024 – May 31, 2025). Forecasts are initialized at 00 and 12 UTC with a lead time of 12 h. Observational profiles are linearly interpolated to ICON model levels ( $\sim$ 13 levels within the target altitude range), yielding approximately 680 000 samples, with ice supersaturation present in  $\sim$ 13 % of cases. Panels show categorical scores for ice supersaturation events: (a) FBI; (b) POD; (c) FPR; (d) precision; (e) MCC; (f) Number of Vaisala RS41 radiosonde RH<sub>ice</sub> event observations.

The POD for ISSR events ( $RH_{ice} > 100$  %) increases from approximately 0.4 for ICON 1-Mom to around 0.6 for ICON 2-Mom, remaining nearly constant across the altitude range in both configurations. For events defined by higher  $RH_{ice}$  thresholds, ICON 2-Mom retains some detection capability, with POD values gradually decreasing to about 0.15—0.2 for  $RH_{ice} > 120$  %. In contrast, consistent with the FBI results, ICON 1-Mom fails to detect  $RH_{ice}$  values above 105 %, yielding POD values near zero throughout the vertical domain.

The FPR remains relatively low across all cases, peaking slightly above 0.1 for ICON 2-Mom at  $RH_{ice} > 100 \%$  (Fig. 4(c)), indicating a limited tendency toward false alarms.

Both schemes yield similar precision values between 0.5 and 0.55 for ISSR events across the entire altitude range (Fig. 4(d)). For higher RH<sub>ice</sub> thresholds, the precision of ICON 2-Mom declines progressively, reaching values as low as 0.2 for RH<sub>ice</sub> > 120 %. In contrast, ICON 1-Mom produces very few or no positive predictions in these regimes, rendering precision largely undefined; accordingly, it is omitted for these cases.

In the context of flight planning, accurate prediction of non-ISSR conditions is equally critical, as false negatives in this category can lead to unnecessary re-routing and, consequently, avoidable increases in CO2 emissions. When treating the complementary events ( $RH_{ice} \leq threshold$ ) as "positive" events, the model exhibits high precision, with average values exceeding 0.9 across all threshold levels. Combined with the low false positive rate observed for  $RH_{ice} > threshold$  events, this high precision underscores the reliability of ICON 2-Mom in correctly identifying non-ISSR conditions.

The MCC shown in Fig. 4(e) summarizes overall classification performance. For ISSR/non-ISSR classification (blue curves), ICON 2-Mom achieves an average MCC of 0.47, indicating moderate predictive skill. In comparison, ICON 1-Mom yields consistently lower values between 0.38 and 0.39. At higher RH<sub>ice</sub> thresholds, the MCC of ICON 2-Mom declines progressively, reaching a minimum of approximately 0.16. In contrast, MCC values for ICON 1-Mom approach zero or become undefined where the numerator vanishes, reflecting a lack of predictive capability in these regimes.

In summary, for ISSR events, ICON 2-Mom achieves a moderate MCC of nearly 0.5 and a POD that is approximately 50 % higher than that of the operational ICON 1-Mom, while maintaining a relatively low FPR below 0.1 across most altitudes. Despite this improvement, a POD of 0.6 indicates that a substantial fraction of events remains undetected. To address this, we continue to investigate potential gains from the ensemble setup introduced in Section 2.2.

# 5.2 Verification of Ensemble Prediction System ICON 2-Mom EPS





We begin with a general evaluation of the ensemble's ability to represent  $RH_{ice}$  variability, using the rank histogram as a diagnostic tool. The rank histogram is constructed by ranking the observed value relative to the ten sorted ensemble forecasts and recording its position across all spatio-temporal matching samples.

Fig. 2(c) shows the resulting histogram for the subset of samples where the observed  $RH_{ice}$  is above 50 %. We consider this restricted rank histogram because ICON tends to underestimate very low humidity values, which are not the subject of this study but would obscure the relevant behavior (also reflected by the  $RH_{ice}$  histogram in Fig. 1(b, bottom)). The histogram exhibits a U-shape, indicating underdispersion, i.e., the ensemble fails to capture the full variability present in the observations. This behavior is partly due to spatial averaging over model grid cells, which tends to smooth out extremes. However, coun-

**Figure 5.** Ensemble verification metrics illustrating the discriminatory skill of the ICON 2-Mom EPS in distinguishing between events and non-events (e.g., ISSR and non-ISSR, shown in blue). The verification period spans 14 months (April 2024 – May 2025), yielding approximately 820 000 samples. (a) Discrimination diagram: Conditional distributions of EPS forecast probabilities, given that the event was observed and not observed in the measurement data. (b) Receiver Operating Characteristic (ROC) curve: POD versus FPR for ice supersaturation events, evaluated across a range of threshold-based decision models derived from the EPS. These pseudo-deterministic models are constructed by applying varying probability thresholds to the ensemble output.

teracting this, so-called upscaling effects of the model tend to display small-scale physical behavior on the model scale. Thus, insufficient parameter perturbations may be another reason, together with the lack of subgrid-scale gravity waves and the use of climatologically prescribed aerosol fields, both of which constrain variability in ice nucleation conditions.

Moreover, the rank histogram reveals a slight negative bias, with observed  $RH_{ice}$  values more often exceeding the ensemble forecast range than falling below it. This suggests a systematic underestimation of  $RH_{ice}$  by the model, at least in parts of the  $RH_{ice} > 50\%$  regime. We found that this mainly occurs at ice supersaturated conditions. However, the rank histogram does not provide any information about magnitudes. Thus, we further analyze the ensemble's ability to classify ISSR and non-ISSR conditions below.

### 5.2.1 ISSR/non-ISSR Discrimination Ability

To evaluate the ensemble's ability to distinguish between ISSR and non-ISSR conditions, we consider the discrimination diagram introduced in Section 4.3. Figure 5(a) shows the conditional distributions of forecast probabilities for observed and non-

observed events (events are defined as  $RH_{ice} > 100$  %, and higher thresholds). For ISSR events, the "not observed" distribution peaks sharply at zero and declines rapidly, indicating strong agreement among ensemble members when no supersaturation is present. In contrast, the "observed" distribution is relatively uniform, suggesting that the ensemble assigns a broad range of probabilities to actual events. As the threshold for supersaturation increases, the "observed" distribution becomes more left-skewed and increasingly overlaps with the "not observed" distribution, indicating a decline in discriminative skill for more extreme events.

To conclude, for the ISSR event, the diagram shows little overlap between the two conditional forecast probability distributions below and above  $\sim$ 0.3, suggesting that a threshold-based conversion of forecast probabilities aimed at classifying ISSR versus non-ISSR may be appropriate.

#### **5.2.2** Threshold-Dependent Performance







As ICON 2-Mom EPS consists of ten ensemble members, ISSR forecast probabilities can be  $0,0.1,0.2,\ldots,1$ . Thus, these values represent the relevant potential thresholds to turn the event forecast probability into an event prediction – yielding classification models as introduced in section 4.3. We refer to these classification models as decision models, with the k-out-of-10 decision model (or simply decision model k) defining the threshold as k/10: if at least k of the 10 ensemble members predict the event, the model outputs a positive prediction:

$$k\text{-out-of-10 decision model}: p_{\text{conv}} = \begin{cases} 1, & \text{if } p \geq \frac{k}{10}, \\ 0, & \text{otherwise}. \end{cases}$$

To evaluate the performance of these pseudo-deterministic decision models, we use the ROC curve (Section 4.3). For the ISSR event, the ROC curve (Fig. 5(b)) shows strong discriminative skill for thresholds of 0.2 and 0.3 (decision models 2 and 3), with POD > 0.8 and FPR < 0.17. A comparison between the scores of the EPS-based decision models and the deterministic ICON 2-Mom model (inset of Fig. 5(b)) shows a substantial improvement in the POD for ISSR events, from approximately 0.6 in the deterministic case to 0.8–0.9 when using ensemble-based decision models. This gain in POD is accompanied by a moderate increase in the FPR, rising from  $\sim$ 0.1 to values between 0.13 and 0.23, depending on the chosen threshold. These results highlight the added value of ensemble forecasts in enhancing event detection or classification.

While the ROC curve provides a comprehensive view of classification performance across thresholds, it treats both classes equally and may obscure performance nuances in the presence of class imbalance. Therefore, we also evaluate the precision–recall (PR) curve, which focuses specifically on the model's performance on the positive class. Similarly to the construction of the ROC curve, the PR curve plots the recall (equivalent to POD) against precision. In Figure 6(a), each EPS-based decision model is represented as a point on the PR curve. The closer the points are to the top right corner, the higher the recall and precision. Although recall remains high even for intermediate thresholds, overall precision is only moderate and deteriorates further for more extreme supersaturation events. This reflects the increasing difficulty of making accurate positive predictions as the event definition becomes more stringent.

We also conduct a complementary analysis by treating the complementary conditions (e.g., non-ISSR) as the positive events, as this perspective is equally relevant for flight routing applications. The PR curve approaches the top right corner, reflecting both high POD and precision, and a zoomed-in view providing details of this region is shown in Appendix D Fig. D1(a).

To conclude this subsection, we shift to a more holistic model assessment using the MCC, as introduced in Section 4.2. The MCC provides a balanced measure of classification skill across both event and non-event categories, making it particularly valuable in the context of imbalanced datasets.

For the ISSR/non-ISSR classification, EPS-based decision models 1–7 consistently outperform their deterministic counterparts (i.e., individual ensemble members), with decision models 3 and 4 achieving the highest MCC values of approximately 0.55. In contrast, the deterministic models yield MCC scores around 0.47 (see Fig. 6(c)). These results reinforce the advantage of ensemble-based decision strategies in capturing both sides of the classification task more effectively.

| $\mathrm{RH}_{ice}$ | max MCC EPS | decision | POD          | FPR          |
|---------------------|-------------|----------|--------------|--------------|
| threshold           | (Det)       | model    |              |              |
| 100%                | 0.55 (0.47) | 3<br>4   | 0.80<br>0.73 | 0.13<br>0.10 |
| 105%                | 0.46 (0.37) | 2<br>3   | 0.77<br>0.68 | 0.14<br>0.11 |
| 110%                | 0.37 (0.28) | 2        | 0.64         | 0.11         |
| 120%                | 0.25 (0.16) | 2        | 0.62         | 0.11         |

**Table 1.** For each  $RH_{ice}$  threshold event, the maximum MCC value of the decision models based on the EPS is shown (rounded to the second decimal place), together with the indices of the corresponding decision model(s). The MCC of the deterministic model (single members) is given in brackets. The last two columns show the ROC values (POD versus FPR) of the decision model(s) with the maximum MCC.

Table 1 summarizes the maximum MCC values achieved for each ice supersaturation threshold, along with the indices of the corresponding EPS-based decision models. For reference, MCC values of the deterministic (single-member) models are shown in brackets. The final columns report the associated POD and FPR values, enabling direct comparison with ROC-based performance. In most cases, the decision models with highest MCC also show favorable POD–FPR combinations, underscoring their robustness across metrics. For the remainder of this study, we focus on ROC-based evaluation using its associated scores, POD and FPR, as a representative framework for assessing decision model performance.

#### 5.2.3 Comparison with IAGOS Data



The RH<sub>ice</sub> density of the IAGOS data, limited to the Northern Hemisphere for better comparison with our radiosonde verification, confirms the characteristic bimodal shape of the RH<sub>ice</sub> density (see inset of Fig. 7). Compared to the ICON data, the first peak in the IAGOS density appears shifted to the right, suggesting fewer near-zero RH<sub>ice</sub> values in the IAGOS dataset than in

Figure 6. Scores that take into account the unbalanced dataset with respect to the ISSR event or higher ice supersaturation events in two different ways: The precision-recall (PR) curve by focusing on the performance of the model with respect to what is defined as the 'positive' event, and the MCC by providing a balanced evaluation measure with respect to all four categories of the confusion matrix. (a) PR curve for the EPS: For increasing prediction probability conversion thresholds, the recall (POD) is plotted against the precision of the corresponding decision model, both with respect to the 'positive' events  $\{RH_{ice} > threshold\}$  (bold crosses) or  $\{RH_{ice} \le threshold\}$  (stars). For the single ensemble members, recall is similarly plotted against precision for both types of events (diamond and thin diamond). A zoom showing the details of the top right corner is provided in the Appendix, Fig. D1. (b) MCC for the EPS decision models as well as for the single members (transparent lines).

the ICON data. The peak around  $RH_{ice} \approx 100$  % is shifted to the left and is less pronounced in the IAGOS data. It also does not reach the same high  $RH_{ice}$  values as ICON.

Nevertheless, up to  $RH_{ice} > 120$  %, the shape of the ROC curves (see Fig. 7) derived from the IAGOS data closely resembles those derived from the radiosonde data (compare Fig. 5(b)). These findings strengthen our verification insights across different, independent observation systems.

# 5.2.4 Longer Forecast Lead Times

Although for many flights 12 hour forecasts are sufficient, we now consider lead time increments from 12 hours up to a maximum of 48 hours, which is the standard time horizon of weather forecasts for flight planning. Figure 8) shows that – as the lead time increases – the ROC curves shift slightly to the right, indicating higher FPR. In contrast, no downward shift of the ROC curves is observed for high POD values of around 0.8 for the first 36 hours and the POD only starts to decrease after

Figure 7. ROC curve of ICON 2-Mom EPS and IAGOS data, the inset figure shows the corresponding  $RH_{ice}$  densities. Evaluation performed with 625 flights from four months (August 2024, October 2024, December 2024, January 2025) on the Northern Hemisphere, leading to  $\sim$ 200 000 spatio-temporal samples.

36 hours. Overall, the degradation is not that severe, and at least up to 36 hours, potential scores remain roughly in the range of POD > 0.8 and FPR < 0.2.

### 450 5.2.5 Incorporating the Ensemble Spread

We further incorporate ensemble spread information in order to get more reliable scores in more specific situations. The ensemble spread should be an indication of the confidence in a forecast and is typically measured by the standard deviation (std). Therefore, in the context of ISSR forecasts, we further differentiate the ROC curve based on the underlying std at each grid point, particularly to achieve a lower FPR.

The inset of Figure 9(a) shows a histogram of the standard deviation of RH<sub>ice</sub>/100%; more than 50 % of the ensemble forecasts have a std below 0.1, with a peak near zero, and only a small proportion have std values greater than 0.2. The colored bins in the histogram serve as a legend for the ROC curves in the main Figure 9(a): The EPS forecasts are partitioned with respect to their std and the corresponding ROC curves are shown in the same color. The trend is consistent with our expectation; the lower the std, the closer the corresponding ROC curve is to the upper left corner, and vice verse, the higher the std, the closer the ROC curve is to the diagonal, indicating that the model has low skill in these cases. In particular, a significantly improved ROC shape is obtained in more than half of the cases – with POD of 0.9-1 and FPR ≤ 0.1 via decision models 1–2. In case the std is greater than 0.1, the ROC curves tend more and more to the diagonal and at least five or six members should

**Figure 8.** ROC curves for increasing forecast lead times and increasing RH<sub>ice</sub> thresholds; time period five months: 1.1.2025–31.5.2025; ICON initial times 0 UTC and 12 UTC; Northern Hemisphere.

indicate ice supersaturation to achieve an FPR of  $\leq 0.1$ . In these cases – depending on the specific std – only a lower POD of 0.3–0.8 can be obtained.

Given the significant variation in ROC curve shapes across different std regimes, we analyze the std values of different  $RH_{ice}$  regimes, particularly when  $RH_{ice}$  is around or above 100 %. In Fig. 9(b), summary statistics of std are shown for increasing 10 % bins of  $RH_{ice}$ . Following an increase in std values, the bins fall before 100 % and reach another local minimum in the  $RH_{ice}$  regime of 100–110 % with a median around 0.1. The relative mean squared error (RMSE) shows a similar qualitative behavior up to  $RH_{ice} < 120$  %. For higher  $RH_{ice}$  regimes, the RMSE increases to its maximum over the whole range of values.




In Fig. 9(c), the full RH<sub>ice</sub> histograms of the observations and the ensemble forecasts are shown, as well as both conditioned on std $\leq$ 0.1; in the case of the observations this is done by assigning the std-value of the corresponding spatio-temporally matched EPS point. For low std-values (std $\leq$ 0.1), the corresponding conditional RH<sub>ice</sub> histograms show a large peak for low humidity values in the same range as the full unconditioned histograms. Another peak is observed for RH<sub>ice</sub> values around 100 %, which is approximately one order of magnitude lower than that of the unconditioned histograms. This difference persists in the supersaturation regime of the histograms, where the maximum RH<sub>ice</sub> values reached in the conditional case are around 130 %, based on the 820 000 verification points (where all counts below 100 were cut in this plot). When comparing the conditional histograms of the model and the observations, the observation histogram exhibits a slightly lower peak around 100 %, similar to the difference observed in the full histograms. In conclusion, even when the model exhibits high confidence, as reflected by a low std, the histogram still displays intermediate supersaturation. This suggests that certain ISSRs can be well predicted.

Figure 9. Event  $RH_{ice} > 100$  %: Inclusion of the ensemble spread of  $RH_{ice}$ , measured by the standard deviation (std) of  $RH_{ice}/100\%$ . (a) ROC curves on sample subsets grouped and color-coded by their standard deviation (std) values. The inset shows the histogram of the std of  $RH_{ice}/100\%$ , which also serves as a legend for the ROC curves corresponding to EPS subsets with associated std. The black ROC represents the original curve based on all samples. (b) Standard deviation and RMSE for 10 % bins of the predicted  $RH_{ice}$  mean; the coral coloured boxes represent the interquartile range (IQR) (middle 50 % of the std data) and the black horizontal line inside the boxes represents the median. The bottom of the box is Q1 (25th percentile) and the top is Q3 (75th percentile). The vertical lines extending from the boxes represent the variability of the data outside Q1 and Q3. They typically reach the minimum and maximum values within  $1.5 \times IQR$ . All data points outside  $1.5 \times IQR$  from Q1 or Q3 are plotted individually as outliers. Blue crosses indicate the RMSE between the ensemble mean and the observed data points. (c) Full histograms of observed and predicted  $RH_{ice}$  values and histograms conditioned on  $std \le 0.1$ ; in the observation case, the corresponding std values were defined by the corresponding spatio-temporally matching EPS values. In the EPS model case, the counts were divided by 10 to obtain a similar range of values to the observations.

The increased predictability in the regime around  $RH_{ice} \approx 100$  % can be explained by a more stable microphysical behavior in this near-thermodynamic equilibrium state, which is captured by the model. In this regime, mature cirrus clouds are dominant compared to young or short-lived cirrus clouds which often form in regions of high ice supersaturation, driven by upward motion from gravity waves or deep convection. These young clouds experience rapid crystal growth due to significant mesoscale temperature fluctuations caused by gravity waves, which create high spatio-temporal variability in supersaturation. The fluctuating vertical motions and ice crystal concentrations make forecasting cloud evolution difficult. As a result, young and short-lived cirrus clouds introduce significant uncertainty in predicting supersaturation, as the microphysical processes are highly dynamic and rapidly changing. In contrast, mature cirrus clouds, approaching thermodynamic equilibrium ( $RH_{ice} \approx 100$ %), display weak supersaturation conditions, typically linked to slow, steady-state ascent. Under these conditions, ice crystals grow and gradually deplete ambient water vapor, creating a balanced system that enhances the predictability of ice crystal evolution and overall cloud dynamics.




In clear-sky regions, where clouds and associated microphysical processes are absent, the predictability of  $RH_{ice}$  is governed primarily by large-scale thermodynamic and dynamical processes. Supersaturation can persist in these regions due to the

lack of ice nuclei. Observations show that clear-sky supersaturation is often associated with weak vertical motions and low temperatures in the upper troposphere, particularly in mid- and high-latitude regions (Kahn et al., 2009). However, mesoscale temperature fluctuations caused by gravity waves can still occur, challenging predictability, particularly for models that do not resolve mesoscale temperature or humidity fluctuations. Overall, while the absence of cloud feedbacks simplifies the microphysical environment, potential variability in temperature, humidity, and vertical motion still introduces uncertainty, i.e., the predictability of RH<sub>ice</sub> in clear skies depends on the given specific large- and mesoscale thermodynamic and dynamical processes.

#### 500 6 Discussion






### 6.1 Observed Standard Deviation of RH<sub>ice</sub>

The results shown in Figures 9(b) and 9(c) share notable similarities with the findings of Borella et al. (2024), who parameterized the subgrid-scale distribution of water vapor in the UTLS using IAGOS data. They observed a predominantly quadratic relationship between the standard deviation of  $RH_{ice}$  and its mean, with a peak occurring between 70 % and approximately 110 %, depending on temperature. Beyond this range, the standard deviation exhibited an upward trend at even higher  $RH_{ice}$  values. Their temperature-dependent analysis further revealed that this peak decreases in magnitude and shifts toward higher  $RH_{ice}$  values as temperature decreases. While our approach to grouping ROC curves by ensemble spread does not currently account for temperature, incorporating it may be a valuable direction for future work.

# 6.2 Comparing Microphysics-Based and Statistical Approaches to Ice Supersaturation

The results of our study demonstrate that the two-moment cloud ice microphysics scheme implemented in ICON provides a microphysically based alternative to prognostic cloud cover schemes – such as the Tompkins scheme used in the IFS model (Tompkins et al., 2007) – that infer ice supersaturation from subgrid-scale humidity distributions. The Tompkins approach offers some advantages for operational weather forecasting due to its computational efficiency and its ability to represent subgrid-scale humidity variability. This can be advantageous for realistic cloud fraction estimates on coarse grids. However, this scheme does not explicitly prognose specific ice mass or ice particle number density, and phase relaxation time is effectively zero because the scheme assumes instantaneous in-cloud equilibrium. Indeed, the current cloud scheme of IFS assumes ice supersaturation only in the cloud-free portion of the grid box (ECMWF, 2024), which can lead to an underestimation or smoothing of ice supersaturation under certain conditions. In contrast, ICON 2-Mom prognoses both specific ice mass and ice particle number density, allowing phase relaxation time to emerge naturally from microphysical relationships. This enables a more direct, microphysics-based simulation of the onset and persistence of ice supersaturation, which is particularly relevant for applications requiring detailed RHi forecasts, such as contrail avoidance. While this approach offers improved physical realism and consistency, it comes with increased computational cost and sensitivity to assumptions about nucleation and particle size distributions. Consequently, careful tuning and validation are necessary, especially in global applications.

# 6.3 Model Resolution and Neighborhood Consideration

Several leading high-resolution NWP models have been validated with respect to RH<sub>ice</sub> using radiosonde data in Thompson et al. (2024). The radiosonde data used were from 2022, covering ten months, and included data from radiosondes of lower or unknown quality than the Vaisala RS41 radiosondes. Model data were interpolated onto radiosonde data, which differs from our approach of interpolating radiosonde data onto model data. The most comparable results are the POD and FPR for  $RH_{ice} > 99.99$  % events, where (POD, FPR) values of (0.46, 0.09) were obtained for the S-WRF model, (0.19, 0.02) for the GFS, and (0.50, 0.10) for the IFS. In all cases, the deterministic model was evaluated.

The study also introduced a 3D neighborhood verification, where the number of ISSR events of horizontal and vertical grid point neighbors affects the identification (definition) of true positives, false positives, false negatives and true negatives. Although in this study neighborhood incorporation is used for model comparison verification, it could also be used to define another meta-model – in this case not based on an EPS model, but on a deterministic NWP model. Of course, a similar definition could also be introduced based on an EPS model. However, although the concept of including neighbors into a model to identify ISSRs is worth exploring, the neighborhood verification presented in the study corresponds to two different models, where the one to be used is individually selected for each radiosonde observation, depending on whether ISSR was actually observed or not. This conditioning on the observation may improve the verification results, as the knowledge of the observation determines the decision of which model to use. For our purpose, which is to define a model for future predictions, it is not appropriate to condition this decision on the observation. But even when including only model neighbor values into a meta-model, the grid resolution we currently use (about 26 km horizontally and about 200–300 m vertically in the height range of interest) may be too low to adequately account for horizontal neighbors. We expect that using a finer grid for ICON predictions may enable such an approach, and most likely improve the overall verification scores.

#### 7 Outlook







# 545 Prediction Improvement via Machine Learning

While the k-out-of-10 decision models are based on intuitive thresholds, they are ultimately heuristic in nature, comparable to a binary classifier trained and validated on model and radiosonde data. Due to the small amount of data ( $\sim$  820 000 samples), we chose to use the gradient boosting tree library CatBoost in classification mode. The results are shown in Appendix Fig. 10. The ROC curve of the CatBoost model shows a slight improvement in the upper left region of interest compared to the k-out-of-10 decision models. In addition, the ROC curve is almost continuous and at high RH<sub>ice</sub> gives access to POD values that are unattainable even for the 1-out-of-10 model, giving a greater degree of control over the desired balance between POD and FPR. Thus, the model reduces the need to run an EPS with many members (but more members slightly improve the predictions; see the 40-member ICON 1-Mom EPS case in Fig. B1). Another advantage of the model is that more features than just RH<sub>ice</sub> itself can easily be added as model inputs. Even extending the feature vector with physical quantities of neighboring cells is equally feasible. The results are very promising and more complex models are being investigated.

**Figure 10.** Comparison of ROC curves of EPS-based CatBoost and EPS-based decision models. CatBoost input features were the RH<sub>ice</sub> values of all ten members. Solid blue ROC curves: Training and validation period from April to December 2024; test data from January to March 2025; ROC calculated for the test data period. Light blue ROC curves: ROC for the EPS-based decision model, evaluated over the test data period. Solid and light orange curves indicate the same setting but with a different training and validation data period (July 2024 to March 2025) and a different test data period (April–June 2024). Except for a larger tree depth of 10, all CatBoost settings were kept at default, and training took about 30 seconds per RH<sub>ice</sub> threshold.

### 8 Conclusions



This study demonstrates the strong potential of EPS-based classification models for ISSR, based on the ICON NWP model enhanced with a two-moment ice microphysics scheme. Compared to ICON 1-Mom, ICON 2-Mom more accurately captures the physical conditions associated with ice supersaturation, where it significantly improves the POD while maintaining a moderate FPR. This improvement is also reflected by the MCC, indicating better overall classification skill.

The EPS model itself, ICON 2-Mom EPS, has served as the foundation for further meta-model developments aimed at constructing deterministic models of ISSR/non-ISSR classification and higher ice supersaturation. These models are designed to provide flight planners with well-scored predictive tools that support actionable decision-making.

Simple k-out-of-10 decision models spanned a wide range of POD-FPR combinations, with many outperforming the deterministic ICON 2-Mom model in terms of POD while maintaining comparable FPRs. For RH<sub>ice</sub> > 100 %, ICON 2-Mom achieves a POD of  $\sim$ 0.6 and an FPR of  $\sim$ 0.1, whereas ICON 2-Mom EPS allows for finer control: decision model 1 yields a POD > 0.9 at an FPR of 0.25, while decision model 9 offers near-zero FPR with reduced POD. This flexibility enables users

to select decision models based on operational cost trade-offs between false positives (e.g., unnecessary diversions) and false negatives (e.g., contrail formation).

Further refinement was achieved by incorporating ensemble spread into the decision making. Grouping ROC curves by the standard deviation of RH<sub>ice</sub> revealed that low-spread conditions correspond to high categorical skill, whereas high-spread scenarios tend toward random performance. This insight was used to define an adaptive meta-model that selects *k* based on ensemble spread, keeping FPR below a target level. This approach relies solely on model data and can be seamlessly integrated into more advanced models.

Building on this statistical framework, a gradient boosting tree classifier was trained as a more sophisticated meta-model. Despite minimal training time and default hyperparameters, it outperformed the *k*-out-of-10 models in the POD–FPR region of interest. Additional advantages include a nearly continuous ROC curve and the ability to incorporate additional features with ease.

While these investigations were ongoing, a contrail avoidance trial based on the ensemble mean of ICON 2-Mom EPS rerouted over 100 flights, demonstrating the operational relevance of this forecasting approach. The results presented here show that EPS-based meta-models bring us closer to reliably identifying conditions conducive to persistent contrail formation. Finally, these findings may inform the European Union's Monitoring, Reporting and Verification (MRV) system, which



uses climate response models to quantify trade-offs between contrails,  $CO_2$  emissions, and other greenhouse gases.  $RH_{ice}$  is a critical input for contrail modeling, yet remains poorly predicted by many operational NWP systems. This study represents a step toward more accurate  $RH_{ice}$  forecasting and improved support for climate-conscious aviation strategies.

Code and data availability. The verification code and data are available under Zenodo (https://doi.org/10.5281/zenodo.15881140).

#### Appendix A: History and Details of the One-Moment Cloud Ice Scheme


The original one-moment scheme is a legacy code developed by Günther Doms at DWD in the 1990s for the COSMO model, which was then known as the Lokalmodell (LM), and operated at a horizontal grid spacing of 7 km (Steppeler et al., 2003). In the 2000s, the same one-moment scheme was used in the operational global model GME, the predecessor of ICON (Majewski et al., 2002). A detailed description of the original one-moment cloud ice scheme is provided in Doms et al. (2021). It shares many similarities with the one-moment schemes by Lin et al. (1983) and Rutledge et al. (1986), both originally developed for mesoscale models.

Over the past 25 years, the operational one-moment cloud ice scheme has undergone many modifications, documented in Section 5.8 of the COSMO 6.0 documentation. Notable updates include warm-rain processes based on Seifert and Beheng (2001), snow particle geometry following Wilson and Ballard (1999), and snow size distributions derived from empirical relationships by Field et al. (2005). Ice crystal concentration is parameterized using the empirical formula by Cooper (1986).

### **Appendix B: Ensemble Verification of ICON 1-Mom EPS**




We also evaluated the ensemble data of the operational ICON 1-Mom EPS with respect to RH<sub>ice</sub>. We wanted to compare the improvement of results such as the POD due to the ensemble setup when the microphysical scheme has not been adapted to a two-moment scheme. Therefore, we considered the ROC curve for the operational 40-member EPS as well as for 10-member subsets, compare Appendix Figure B1. By similarly defining decision models for ISSR, the POD can be increased to more than 0.8 with an FPR remaining below 0.2, which holds true for both the 40- and 10-member EPS. The full EPS yields a more fine-grained curve with slightly higher POD values in the top left corner than the 10-member EPS. Overall, the potential of an ensemble is highlighted in both cases, especially with respect to a possible increase in POD. However, the operational 1-Mom EPS still fails to predict events with higher RH<sub>ice</sub> values (see inset in Fig. B1), as it relies on an NWP model with insufficient physical parameterization for larger RH<sub>ice</sub> values. This finding aligns with studies that emphasize the importance of model quality as a key factor in the success of ensemble prediction systems (Wang et al., 2018; Du et al., 2018).

Finally, we wanted to confirm that the specific selection of ten members from the original 40 had little or no effect on the scores due to the way the ensemble is generated. Therefore, we performed a 10-out-of-40 bootstrap and considered the mean and standard deviation of the corresponding points of the ROC curves of each subset EPS. The resulting standard deviation is negligibly small, encouraging us to transfer this finding to our ICON 2-Mom EPS, using the first ten members.

**Figure B1.** ROC curves for the ICON 2-Mom EPS (orange), the operational ICON 1-Mom EPS with 40 members (blue) and for the corresponding ICON 1-Mom EPS subsets with 10 members (green). For the latter, we randomly selected 1000 10-member EPS subsets, calculated the ROC curve for each and plotted the mean and standard deviation of the corresponding points on the curve. The inset figure shows ROC curves for the ICON 1-Mom 40-member EPS for higher RH<sub>ice</sub> thresholds up to 106 %. Evaluation for three months (August 2024, October 2024, January 2025); ICON initial times 0 and 12 UTC; ICON forecast lead time 12 h; Northern Hemisphere.

### **Appendix C: Calculation of RH**<sub>ice</sub>

# C1 Computation of RH<sub>ice</sub> for radiosonde data

In the TEMP BUFR files, as disseminated through the Global Telecommunication System (GTS), the dew point temperature  $(T_d)$  is provided, which allows us to compute the water vapor partial pressure (e) using the formula from Hardy (1998), ensuring consistency with the processing applied by radiosonde manufacturers, such as Vaisala. We further calculate the saturation vapor pressure over ice  $(e_i)$  consistently with the formula used in ICON which is given by

$$e_i = b_1 \frac{\exp(b_{2i}(T - b_3))}{T - b_{4i}}$$
 (C1)

with coefficients

$$b_1 = 610.78, b_{2i} = 21.87, b_3 = 273.16, b_{4i} = 7.66.$$

and referred to as the Magnus-Tetens-Murray approximation (Magnus, 1844; Tetens, 1930; Murray, 1967). Therewith, we receive

$$RH_{ice} = \frac{e}{e_i} 100 \%.$$
 (C2)

# 625 C2 Computation of RH<sub>ice</sub> for ICON Data

First we calculate the water vapor partial pressure e by

$$e = r_v T \rho q v$$
,


where the temperature T (in K), the density of moist air  $\rho$  (in kg/m<sup>3</sup>), and the specific water vapor content qv (in kg/kg) are output variables of ICON, and  $r_v = 461.51$  is the gas constant for water vapor. Finally, we calculate  $e_i$  again with C1 and RH<sub>ice</sub> with C2.

Note that recently, as of May 2025, the coefficients in the C1 formula for the saturation vapor pressure over ice in the operational ICON model have been updated. We still use the old version of the coefficients given in C1 in our dedicated system and therefore in our verification analysis. However, at -37 °C, the error is only about 2 %.

# C3 Computation of RHice for IAGOS Data

In the IAGOS NRT dataset, RH<sub>ice</sub> is already included and has been calculated using the formula from Sonntag (1994), which is very similar to the Hardy formula.

# Appendix D: Details of Precision-Recall Curve for non-ISSR

In Fig. D1 a zoom of the top right of Fig. 6 is provided, where the details of the PR curve for the non-ISSR event and for the events  $\{RH_{ice} \leq threshold\}$  with threshold in  $\{105\%, 110\%, 120\%\}$  can be seen. Note that *decision model* k here refers to the decision model which requires at least k ensemble members with the event  $\{RH_{ice} \leq threshold\}$ .

Figure D1. Precision-recall curve of events  $\{RH_{ice} \leq threshold\}$  with threshold in  $\{100\%, 105\%, 110\%, 120\%\}$  (zoom of top right of Figure 6 with adapted markers on the lines). Markers on the lines indicate the scores corresponding to the decision models based on the EPS. Thin diamonds inidcate the scores of the single ensemble members.

# Appendix E: Binary Classification Models: CatBoost

CatBoost is a machine learning library based on gradient boosting on decision trees, where input features are either real values or categorical values. Prediction can happen either as regression or classification. For the task at hand, CatBoost was used in classification mode, with the cross-entropy loss J used for training:

$$J(\boldsymbol{y}, \boldsymbol{p}) = -\frac{1}{N} \sum_{i=1}^{N} [y_i \log(p_i) + (1 - y_i) \log(1 - p_i)]$$


where N is the total number of samples (spatio-temporal matching points of model and observation),  $y_i$  is 1 if an event was observed, otherwise 0, and  $p_i$  is the prediction probability of the model. The samples were divided into 75 % training and validation data and 25 % test data. The test data were taken from different months than the training/validation data to minimize the effect of potential correlations in the data. Figure 10 shows the performance of the model on the test data, compared to the EPS-based decision models of this study applied to the test data period.

Author contributions. LS and AS supervised and conceived the study, with contributions from MH and CGK. AS implemented the adapted two-moment ice microphysics scheme in ICON, based on a previous COSMO/GME implementation by CGK, and set up the dedicated ICON system at DWD. MH developed the verification methods and conducted the analysis. All authors discussed the results. MH drafted the manuscript with input from all authors, and all contributed to the final version.

Competing interests. The authors declare that they have no competing interests.




Acknowledgements. MH and CGK are funded by the German Federal Ministry for Economic Affairs and Climate Action (BMWK) under the German Aerospace Research Programme (LuFo) (FKZ: 20M2111F).

The authors thank the D-KULT project partners, especially Klaus Gierens, for helpful discussions. The authors also thank Prof. Ulrich Schumann for insightful discussions and sharing knowledge on the topic. Ruud Dirksen is acknowledged for providing valuable insights on radiosonde measurements.

Further thanks go to John Walter Acevedo Valencia for introducing MH to IAGOS data and to Susanne Rohs and Damien Boulanger for their generous support on IAGOS data and for providing the IAGOS NRT data. IAGOS data were created with support from the European Commission, national agencies in Germany (BMBF), France (MESR), and the UK (NERC), and the IAGOS member institutions (http://www.iagos.org/partners). The participating airlines (Lufthansa, Air France, Austrian, China Airlines, Hawaiian Airlines, Air Canada, Iberia, Eurowings Discover, Cathay Pacific, Air Namibia, Sabena) supported IAGOS by carrying the measurement equipment free of charge since 1994. The data are available at http://www.iagos.fr thanks to additional support from AERIS.

The authors thank Björn Beckmann for fruitful discussions on detailing the dedicated system, Thomas Hanisch and Tobias Göcke for helping to set it up, and Sven Ulbrich for curating its data. Christoph Gebhardt, Chiara Marsigli and Jochen Förstner are acknowledged for providing details on the ensemble generation at DWD. MH especially thanks Felix Reinhardt for fruitful discussions on the verification implementation and insightful discussions on the results.

The two anonymous referees are acknowledged for their constructive comments which improved the final manuscript.

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
