# Peer review of "Predicting Ice Supersaturation for Contrail Avoidance: Ensemble Forecasting using ICON with Two-Moment Ice Microphysics"

_EGUsphere, 2025_

## Referee Comment (RC1)

**Comments on "Predicting Ice Supersaturation for Contrail Avoidance: Ensemble Forecasting using ICON with Two-Moment Ice Microphysics" by Maleen Hanst et al.**
**https://doi.org/10.5194/egusphere-2025-3312**

The authors have done important work. While earlier studies attempted to apply correction methods to better represent ice-supersaturated regions (ISSRs), this study aims to implement an improved two-moment ice microphysics scheme within the ICON model framework.

An improved representation of supersaturation in models is essential for accurately forecasting cirrus and, in particular, the potential for contrail formation. Furthermore, a better representation of ice supersaturation and contrail estimation is fundamental for estimating the radiative forcing of cirrus and contrails, enabling flight re-routing and more realistic modeling of the water budget. The proposed two-moment scheme is an important step forward in the development of the ICON model.

In addition to the modified two-moment scheme, this study estimates the benefits of ensemble member simulations for ISSR prediction and uncertainty estimation. Simulated temperature and relative humidity from the operational ICON one-moment ice microphysics scheme and the modified two-moment schemes are compared with radiosonde measurements and, to a lesser extent, IAGOS measurements. The comparison shows that ICON's representation of ice supersaturation improves significantly, when using the two-moment schemes.

To make the results more accessible to a broader community, the structure of the text needs improvement. Please try to get quicker to the point. Restructuring the entire text is also necessary to avoid repetition and establish a logical structure. The structure of a paper should guide readers step by step through the topic and help them understand the subject matter. Please see below for more details.

A possible structure for the beginning of the manuscript could be:

1. Introduction
2. Data
2.1 Model
2.1.1 Operational ICON 1-Mom
2.1.2 Modified ICON 2-Mom and ICON 2-MOM EPS
2.2 Measurements
2.2.1 Radiosonde
- processing
- data extraction
2.2.2 IAGOS
- resolution, uncertainty
- processing and extraction

3. Methods
3.1 processing of model data
3.2 Validation scores
......
This structure is only a suggestion, and I will leave it up to the authors to decide how to restructure their text.

A concern along the same line. The paper includes ten different metrics, some of which are not well introduced. You may consider focusing on fewer and the most important metrics. For details please see my major and minor comments. There are also many inconsistent abbreviations and mathematical formulas that must be homogenized to avoid ambiguity.

Most of my comments are suggestions, and I hope the authors find them helpful and do not misinterpret them.

**Major comments:**

- The text can be shortened and made more concise by removing some parts or simply by restructuring paragraphs. For example, L197-200: "Again, the humidity measurement technology used here combines humidity and temperature sensing elements. In more detail, it consists of a capacitive relative humidity sensor (Humicap-H, Vaisala, Finland) and a platinum resistance sensor (PT100) for the measurement of the temperature at the humidity sensing surface. " This could be shortened:
  "The temperature is measured using a platinum resistance sensor (PT100) and rel. humidity is measured using capacitive relative humidity sensor (Humicap-H, Vaisala, Finland)."
  I do understand that everyone has their own writing style, but since the paper is already quite long, writing concisely will make the content more accessible to potential readers.

  Another example is the repeated mentioning of information, like "reducing the number of ensemble members". This point is first introduced in L122, and then revisited in L146–147 and L166–168.

- Since you are introducing an improved version of the two-moment scheme you may spend a little more time one the actual improvement. Would it be worth to shift it from Appendix A to the main text?

- The analysis makes use of ten different "metrics": confusion matrix, FBI, POD, FPR, precision, MCC, ROC, k-out-of-10, Youden index, and F1. You may want to consider reducing the number of metrics and focusing on the most important ones that can be used throughout the paper. If you decide to keep them all, they must be introduced well, either in the methods section (preferred) or in the text when they are first used. Please provide the possible technical ranges of the indices/metrics, common values, and the desired value so that the reader can interpret the metrics.

- The discussion session reads more like a summary. If introducing a dedicated discussion session, then the results of the analysis should discussed by setting them into context with existing literature. For example to discuss differences or similarities in distributions of ISSR and to explain the causes for potential differences.

**Minor comments:**

- The manuscript focuses specifically on contrail formation. However, the formation of cirrus clouds and supersaturation is also relevant to the water budget in the upper troposphere and

lower stratosphere. You could mention this as an additional motivation for improving the representation of supersaturation.

- L40: Could you provide a reference for interested readers?

- L41: NWP was already introduced in Line 35

- L43: RH_ice was already introduced in Line 35

- L41-47: These lines contain redundant information by mentioning the uncertainty and lack of in situ observations multiple times. Please revise accordingly.

- L52: Maybe "...and returns adjusted values of RH_ice?" instead of "outputs".

- L52: "Wang et al. (2025)...." Please note that ERA5 is suspected to be biased in terms of relative humidity, which causes problems in resolving ISSR. This is due to the fact that RH_ice is clipped to a maximum value, as well as the spatial resolution of the model grid. However, there is no consensus on whether RH_ice is too low or too high at the tropopause level. Several correction methods for ERA5 have been provided, e.g.,

  1) Schumann, U. and Graf, K.: Aviation-induced cirrus and radiation changes at diurnal timescales, J. Geophys. Res.-Atmos., 118, 2404–2421, https://doi.org/10.1002/jgrd.50184, 2013.

  2) Schumann, U., Penner, J. E., Chen, Y., Zhou, C., and Graf, K.: Dehydration effects from contrails in a coupled contrail--climate model, Atmos. Chem. Phys., 15, 11179–11199, https://doi.org/10.5194/acp-15-11179-2015, 2015.

  3) Teoh, R., Schumann, U., Gryspeerdt, E., Shapiro, M., Molloy, J., Koudis, G., Voigt, C., and Stettler, M. E. J.: Aviation contrail climate effects in the North Atlantic from 2016 to 2021, Atmos. Chem. Phys., 22, 10919–10935, https://doi.org/10.5194/acp-22-10919-2022, 2022a

- L54: At the end of the sentence: "...when validated against test data"?

- L84: NWP already introduced

- L92: Throughout the manuscript, there are multiple versions of RH_ice. There is: RHice and RHice. Figures use RHi. All of these combinations are sometimes in italics and sometimes not. Choose one version and stick to it throughout the script.

- L96: What are the typical challenges? They are known to you, but since you are making this point, please briefly mention them. Or are they given in the next sentences? If so, I would suggest to write, ... with forecast applications, such as data assimilation and model uncertainties, the interpretation of the resulting forecast..."

- L126: abbreviate "relative humidity of ice"?

- L140: and elsewhere. To facilitate understanding of the paper, clearly define your different model setups and provide unique abbreviations for each configuration. ICON (for the operational 1-Moment scheme), ICON 2-Mom (for the new 2-Moment scheme), and ICON EPS 2-Mom. Then, stick to these abbreviations. The current version uses various

abbreviations and paraphrases. Sometimes, the 2-Mom model is also called the deterministic model. These variations make it difficult to understand the paper's content and cause unnecessary ambiguities.

- L164 and elsewhere: For ranges, e.g., 8.5-12.5, use spaced en-dashes "--" see ACP style guide

- 169 - 170: Why is ICON 2-Mom now cursive? Please see comment above.

- L184: What does "TEMP BUFR" mean? Enter the full name here. Explaining BUFR in the next sentence is insufficient and too late.

- L185: It is usually first the long name, followed by the abbreviation in brackets.

- Fig2: The subpanels and labels are small but still legible. Would it not make sense to split the plot and place the individual plots in the positions where they are discussed? This would also prevent flipping back and forth through multiple pages when referencing back from Section 4. The authors may want to consider this.

- L209: Please explain which model you mean with "dedicated model". ICON 1-Mom, 2-Mom?

- L213: Call it ICON 2-mom?

- L215: Why ICON 1-mom now in italics?

- L224: Is the new paragraph required?

- L224: Is "density tail" the correct term? I have not found it in the literature. Would you call it the "tail of the density distribution"?

- L224: "operational system"? you mean ICON 1-mom?

- L230: "The ICON grid employed features a horizon..." Please check the grammar of this sentence.

- L232-233: Why do you use the closest match to the radiosonde station instead of the actual position of the radiosonde? Radiosondes drift horizontally by several kilometers and may end up in a different grid cell.

- Sec 4.1.1 and 4.1.2: What are the fundamental differences between these subsections? Would it not be better to start with 4.1.2 and explain how the data is extracted, and then analyze and compare the data? Additionally, both could be done in one subsection ending in a less fragmented text.

- L241: "An ICON spin up time of a minimum of 6 hours was required" This does not fit here. Would it be better placed in the introduction of the ICON model?

- L247: "...simple scatter plot." Please explicitly mention which plot you are referring to so that the reader can more easily identify it. In my opinion, showing it at the end of the next sentence is too late. I found myself wondering where to look.

- L262: Why is the first part in brackets and the rest in the subscript?

- L265: You may state the possible range of FBI and that a value close to 1 would be desirable.

- L277: "...the POD increases from about 0.4 for ICON 1-Mom..." I guess this is for the 100% threshold? You should mention that.

- L282: "..., also known as sensitivity,..." Does this refer to the probability of detection? If so, please introduce the term "sensitivity" when defining the POD.

- L232: "specificity" is not defined. Do you mean sensitivity?

- L301: "Another way..." if I'm not mistaken, no method has been mentioned so far to address the imbalance. At least, none has been mentioned explicitly. Please check and revise the script accordingly.

- L306 here and elsewhere: minus signs with $-$.

- L321-322: You may also mention that humidity varies greatly in the atmosphere, and radiosondes have a much higher spatial and temporal resolution than models.

- L324-327: What do you mean by "post-processing"? Please specify. As I understand it, you are proposing a new two-moment scheme that allows for supersaturation. Your previous analysis showed that the two-moment scheme performs better than the one-moment scheme. Does the new two-moment scheme just need more refinement or adjustment instead of additional post-processing steps? Alternatively, please explain what is meant by "post-processing."

- L329-330: " distinguish between ISSR and non-ISSR conditions (or higher supersaturation)," Should it be "non-ISSR and ISSR (or higher supersaturation) conditions"?

- L330-332: Why is non-ISSR not an option? As I understand it, the critical point is non-ISSR versus ISSR. At some point, the degree of supersaturation might no longer be relevant in determining whether a contrail can form, but rather, the duration that the ISSR and a potential contrail persist.

- Fig4 (b) has no title. Since you provided one for the inserted plot, you should also provide one for the main graph.

- L345: Would it be appropriate to start the following paragraphs with a new heading, such as: "Categorical Scores of ICON 2-Mom ENS". You mentioned investigating the continuous values first and then switching to the metrics. This would also keep it consistent with section 4.1. It's just a suggestion.

- Figure 5 goes over half of the page. You may want to reduce its size and incorporate some of the explanations into the text.

- Fig 7 : Please explain "ROC stratification". This has not been done in the text.

- L444: Earlier, you used a space "\," between the number and the percent sign. From the ACP Submission Guidelines: Spaces must be included between number and unit (e.g. 1 %, 1 m). Please revise the manuscript accordingly.

- L452-456: "When comparing [...] In conclusion, even when the model exhibits high confidence, as reflected by a low standard deviation, the histogram still displays intermediate supersaturation. This suggests that certain ISSRs can be well predicted."
I have a difficult time following this line of reasoning. Please explain better and potentially

rewrite. Do you mean that the peak around RH_ice = 100 % is not fully resolved and only closely resembles the radiosonde observations?
Please define what you mean by "certain." Does this refer to cases with high, medium/intermediate, or low supersaturation? Which ISSR would be missed and which are well represented?

- L461: What is the ROC stratification approach? Stratification did not appear before, only in the caption of Fig. 7, which is insufficient.

- L467: I would avoid using the abbreviation "MTFs." It is only used one more time.

- L483: Again, it's just one paragraph in a new subsection. Couldn't it be discussed together with the radiosonde observations? A more important question: Do you need the comparison with the IAGOS data at all? What additional information does 4.2.4 provide compared to the radio-soundings? If you want to take advantage of the coverage over the Oceans and at flight levels, where most of the commercial aircraft are operating, then you should spend more time on this analysis and explain this.  But subsection 4.2.4  is very brief and does not provide new conclusions

- L488: "...radiosonde data (Fig.8)" may suggest that the plot contains radiosonde data, but none is plotted. Perhaps rephrase. Alternatively, add the radiosonde data to the plot. This may make a visual comparison easier.

- L662: units not in italics

- In the entire reference section the DOIs are missing.

---

## Referee Comment (RC2)

**Comment "Predicting Ice Supersaturation for Contrail Avoidance: Ensemble Forecasting using ICON with Two-Moment Ice Microphysics" by Hanst et al.**

**General Comments**

The study presents an improved Two-Moment microphysical scheme and then evaluates the performance of the ICON model equipped with this scheme in predicting ice supersaturation up to 48 hours in advance. The evaluation is carried out using observation data measured by radiosonde and aircraft as references. The performance of the model with this new parameterization is also compared to that of the operational One-Moment version of the scheme.

The study is part of efforts to improve flight routing in order to avoid areas of ice supersaturation. Assessment scores that are particularly relevant and well suited to this objective were used. The results show that the Two-Moment scheme provides better performance than One-Moment one for ISSR forecasts. The authors also explored a machine learning approach, which proved promising. This work is of high quality and importance, and it represents a significant contribution to ISSR forecasting. The paper fits the scope of ACP. I highly recommend its publication, but some improvements are necessary.

Overall, the manuscript could benefit from a more concise presentation. Some statements are repeated throughout the text, and certain elements currently included in the Introduction would be more appropriately placed in the Model setup Section. In addition, a significant issue remains concerning the evaluation of the model against fine-scale observations used as a reference, as well as the treatment of the uncertainty associated with these observations.

**Specific Major Comments**

1. The introduction could be presented more concisely:

   a. Lines 69 to 100 should be limited to presenting the ensemble forecasting system as described, and include a bit more details on the one-moment scheme, mentioning its limitations, and then explaining the motivation for transitioning to a two-mode scheme.

   b. The sentence spanning lines 69–77 should be moved to the end of the Introduction; otherwise, the aims of the work are introduced too early.

   c. The paragraph from lines 80 to 82 should be moved to the Model *Setup* section.

   d. The paragraph from lines 101 to 104 should be merged with the last paragraph of the introduction, for example as follows: "This work consists of presenting a new version of the scheme … and assessing its performance against observations and the old version. It is structured as follows: …".

2. Description of the parameterization: The appendix (lines 609–610) indicates that this parameterization is a simplified version of that of Köhler and Seifert (2015), but it does not specify how it differs, and this is also not clarified in Section 2.1. I suggest summarizing Sections A1, A2, and A3 – within the main body of the article in Section 2.1, to improve readability, understanding, and reproducibility, especially since this parameterization does not appear to have been published elsewhere. This could be achieved without significantly lengthening the manuscript.

3. The presentation of the evaluation metrics is currently scattered throughout the *Results* section, which somewhat affects the readability. It may be clearer and more convenient for the reader if the authors dedicate a specific section to these metrics and concisely.

4. Line 154: You mention that "the system includes stochastic perturbations of selected parameterisations". Could you specify which parameterisations are perturbed in your simulations? Furthermore, given that the model may become unrealistic with certain parameter values, could you clarify how the stochastic perturbations are applied? Is it limited to an area of the space of the parameter values, relevant to ISSR forecasting, in order to ensure that the model produces realistic values for these forecasts?

5. Section 4.1.2 – from line 230 to line 235: You mentioned that, in order to perform the spatio-temporal comparison, the observations (radiosonde and IAGOS data) were made comparable to the model grid by vertically interpolating the observations to the model levels. Unless I am mistaken, no horizontal spatialization was performed, only colocation. Consequently, these vertically interpolated observation values can be considered as relatively local.

  a. Could you discuss the relevance of their approach of comparing these vertically interpolated values with those of the model, which has a horizontal resolution of 26 km?

  b. Communities working on precipitation generally address resolution differences using spatial kriging, but this requires closer stations in order to calculate spatial covariance. I imagine that this type of approach is not usable in your study because measurements are not made everywhere at the same time and the distance between measurement points is very large. Consequently, it is important to discuss the implications of the differences in resolution on the results.

6. Ligne 258: In connection with the previous comment, is it appropriate to systematically use the threshold 100 % to define ice supersaturation in the model? Given the model's horizontal resolution (26 km), local areas of supersaturation may exist even when the grid-cell average remains below the saturation threshold. Have you tested the impact of using a slightly lower threshold, for example between 90 % and 100 %, to define ISSRs in the model, on its performance?

7. In the description of the observations, an uncertainty is mentioned, but it is not discussed further. Could the authors consider taking this uncertainty into account in the evaluation, as they did in Section 4.2.3 for the model by incorporating its ensemble spread? In other words, have you examined whether adjusting the threshold according to observations' uncertainties range, could impact the model performance score?

8. In lines 324-325: the authors state: "We also observe a more pronounced negative bias within the rank histogram, indicating that the model tends to underestimate RHice more often than it overestimates RHice." Could this observation not be explained, at least in part, by the difference in spatial resolution between the observations and the model?

9. Ligne 616 : The mean diameter of ice crystals detrained from deep convection has been set to 200 μm. Is this choice based on observations? If not, would it not be relevant to consider it as a tuning parameter by defining a plausible range of values?

10. Lignes 629-630 : The authors assume that the concentration of desert aerosols (Ndus) is constant at 200 hPa and set to 1000 m⁻³. Is this assumption realistic in all regions, particularly in tropical deep convection areas, where aerosol vertical distributions may vary due to convective transport processes? Wouldn't it be appropriate to set Ndus based on latitude and/or treat it as a parameter to be explored within a plausible range of values?

11. Section 5.3: The authors discuss the model resolution and neighborhood considerations. In connection with Comment #2, could the authors comment on the uncertainties introduced in the results by the difference in resolution?

**Specific Minor Comments**

1. Line 2. ISSR should be defined earlier as RHi>100/%, preferably in the first sentence rather than in line 6.

2. Line 23: Please specify the phenomena to which you are referring.

3. Line 59: The part of the sentence from "evaluated..." to "were found" is not very easy to read, as it presents metrics without explaining their implications and also uses acronyms without definitions. Since these details are not essential for understanding the rest of the paragraph,, I suggest that the authors replace this part with a more general formulation such as: *"evaluated using accuracy assessment metrics."*

4. In the caption of Fig. 1, please specify the altitude or pressure level corresponding to what you define as the near tropopause.

5. The results presented for the machine learning approach are very interesting. However, they could be better highlighted in a separate article where the method would be described in more detail, making it accessible and useful to a wider audience.

---

## Author Comment (AC1)

**Response to the Anonymous Reviewers of**

"Predicting Ice Supersaturation for Contrail Avoidance: Ensemble Forecasting using ICON with Two-Moment Ice Microphysics"

by M. Hanst, C.G. Köhler, A. Seifert, and L. Schlemmer.

October 27, 2025

**General Response**

We thank both reviewers for their valuable comments and suggestions. We have carefully revised the manuscript in response to their feedback and provide detailed point—by—point responses below.

In particular, following the Referees' recommendations regarding structure and length, we have streamlined the manuscript by removing redundant contents and improving clarity. A key structural change is the introduction of a dedicated "Verification Methods" section, which now precedes and supports the renamed and revised "Verification Results" section (former "Verification Analysis").

As a result of these revisions, the overall manuscript length has been reduced from 684 to 654 lines (excluding the Acknowledgements and bibliography). While the reduction in length is modest, it is worth noting that we now address additional points raised during the review. More importantly, we are confident that the manuscript's structure and conciseness have been significantly improved.

**Overview of Main Changes**

In accordance with the suggestions from both Referees, we have implemented the following key revisions:

- Restructured the entire manuscript and in particular the second half of the "Introduction" to improve readability and focus.
- Moved the details of the two-moment ice microphysics parameterization from Appendix A into the main text ("Model" section).
- Introduced a dedicated "Verification Methods" section to consolidate and clarify the evaluation metrics.
- Restructured and slightly expanded the dedicated "Discussion" section to better highlight comparisons with related studies and the broader implications of our findings.
- Added a concise "Outlook" section presenting our initial machine learning results, emphasizing their potential for future research.

**Additional Changes**

In addition to the revisions made in direct response to the Referees' comments, we have implemented several further improvements to enhance the clarity and structure of the manuscript:

- A brief paragraph was added to the "Introduction" and a dedicated subsection in the revised "Discussion" to highlight the principles of the Tompkins cloud cover scheme, particularly in contrast to ICON's microphysics-based approach.
- Several passages throughout the manuscript were rephrased to improve clarity, streamline the text, and enhance overall readability.
- All modifications in the revised manuscript are marked in red for transparency, including cases where mainly the placement of paragraphs was adjusted.

**Response to Anonymous Reviewer #1**

**General comments**

The authors have done important work. While earlier studies attempted to apply correction methods to better represent ice-supersaturated regions (ISSRs), this study aims to implement an improved two-moment ice microphysics scheme within the ICON model framework.

An improved representation of supersaturation in models is essential for accurately forecasting cirrus and, in particular, the potential for contrail formation. Furthermore, a better representation of ice supersaturation and contrail estimation is fundamental for estimating the radiative forcing of cirrus and contrails, enabling flight re-routing and more realistic modeling of the water budget. The proposed two-moment scheme is an important step forward in the development of the ICON model.

In addition to the modified two-moment scheme, this study estimates the benefits of ensemble member simulations for ISSR prediction and uncertainty estimation. Simulated temperature and relative humidity from the operational ICON one-moment ice microphysics scheme and themodified two-moment schemes are compared with radiosonde measurements and, to a lesser extent, IAGOS measurements. The comparison shows that ICON's representation of ice supersaturation improves significantly, when using the two-moment schemes.

To make the results more accessible to a broader community, the structure of the text needs improvement. Please try to get quicker to the point. Restructuring the entire text is also necessary to avoid repetition and establish a logical structure. The structure of a paper should guide readers step by step through the topic and help them understand the subject matter. Please see below for more details.

A possible structure for the beginning of the manuscript could be:

- 1. Introduction
- 2. Data
- 2.1 Model
- 2.1.1 Operational ICON 1-Mom
- 2.1.2 Modified ICON 2-Mom and ICON 2-MOM EPS
- 2.2 Measurements
- 2.2.1 Radiosonde
- processing
- data extraction
- 2.2.2 IAGOS
- resolution, uncertainty
- processing and extraction
- 3. Methods
- 3.1 processing of model data
- 3.2 Validation scores

. . . . .

This structure is only a suggestion, and I will leave it up to the authors to decide how to restructure their text.

A concern along the same line. The paper includes ten different metrics, some of which are not well introduced. You may consider focusing on fewer and the most important metrics. For details please see my major and minor comments. There are also many inconsistent abbreviations and mathematical formulas that must be homogenized to avoid ambiguity.

Most of my comments are suggestions, and I hope the authors find them helpful and do not misinterpret them.

**Response**

We acknowledge the Referee for the positive and constructive feedback, and we are committed to implementing their recommendations. In response, we have removed redundancies and restructured the manuscript, incorporating several of your helpful suggestions.

We maintain the distinction between "Model" and "Observation Data", reflected in two dedicated main sections. As suggested, we now include a brief description of the 1-moment scheme at the beginning of the Model section and refer to the Appendix for further details. The 2-moment scheme is explained in more depth in the main text, along with additional information on ensemble generation – both adjustments are also in line with the recommendations of Referee #2.

The Observation Data section has been streamlined, and the data processing steps are now placed at the beginning of the newly introduced "Verification Methods" section (in the original manuscript they appeared in the "Verification Analysis" section), as they are directly linked to the matching with model data.

The new "Verification Methods" section – introduced in response to this comment and also supported by Referee #2 – explains: (1) the spatio-temporal matching of model and observational data, (2) the categorical metrics used for deterministic model evaluation and as the basis for ensemble evaluation, and (3) the methods applied for categorical verification of the ensemble.

Our former "Verification Analysis" section has been split into this new "Verification Methods" section and a revised "Verification Results" section, which has been accordingly shortened and slightly restructured for improved clarity.

Finally, the preliminary machine learning results are now presented in a concise "Outlook" section to clearly separate this distinct approach from the standard verification framework.

**Adapted manuscript structure**

- 1 Introduction
- 2 Model
  - 2.1 Two–Moment Cloud Ice Microphysics Parameterization in ICON
    - \* Deep Moist Convection
    - \* Homogeneous Ice Nucleation
    - \* Heterogeneous Ice Nucleation
  - 2.2 Ensemble Generation
  - 2.3 Model Setup
- 3 Observation Data
  - 3.1 Vaisala RS41 Radiosonde Data
  - 3.2 IAGOS Near-Real-Time Data
- 4 Verification Methods
  - 4.1 Spatio-Temporal Matching of Model and Observation Data
  - 4.2 Categorical Metrics
  - 4.2 Categorical Verification of Probabilistic Model
    - \* Discrimination diagram
    - \* Receiver Operating Characteristic Curve (ROC)
- 5 Verification Results
  - 5.1 Verification of Deterministic Model ICON 2–Mom
    - \* 5.1.1 Relative Frequency Distribution of RHice
    - \* 5.1.2 Continuous Spatio-Temporal Comparison
    - \* 5.1.3 Categorical Verification
  - 5.2 Verification of Ensemble Prediction System ICON 2–Mom EPS
    - \* 5.2.1 ISSR/non-ISSR Discrimination Ability
    - \* 5.2.2 Threshold-Dependent Performance
    - \* 5.2.3 Comparison with IAGOS Data
    - \* 5.2.4 Longer Forecast Lead Times
    - \* 5.2.5 Incorporating the Ensemble Spread
- 6 Discussion
  - 6.1 Observed Standard Deviation of RHice
  - 6.2 Comparing Microphysics-Based and Statistical Approaches to Ice Supersaturation
  - 6.3 Model Resolution and Neighborhood Consideration
- 7 Outlook
  - \* Prediction Improvement via Machine Learning
- 8 Conclusion

**Major comments**

• Comment 1: The text can be shortened and made more concise by removing some parts or simply by restructuring paragraphs. For example, L197-200: "Again, the humidity measurement technology used here combines humidity and temperature sensing elements. In more detail, it consists of a capacitive relative humidity sensor (Humicap-H, Vaisala, Finland) and a platinum resistance sensor (PT100) for the measurement of the temperature at the humidity sensing surface." This could be shortened: "The temperature is measured using a platinum resistance sensor (PT100) and rel. humidity is measured using capacitive relative humidity sensor (Humicap-H, Vaisala, Finland)." I do understand that everyone has their own writing style, but since the paper is already quite long, writing concisely will make the content more accessible to potential readers. Another example is the repeated mentioning of information, like "reducing the number of ensemble members". This point is first introduced in L122, and then revisited in L146–147 and L166–168.

**Response**

- We appreciate the suggestion to streamline the text. Accordingly, we shortened the description. Regarding the first example, we agree that the detailed description of the measurement techniques may not be essential for our context. Since interested readers are already directed to relevant sources for further information—such as Vaisala (2013) and the IAGOS instrumentation overview (https://iagos.aerisdata.fr/instrumentation) we have removed the explicit descriptions for both the radiosonde and IAGOS cases.
- We agree that repeatedly mentioning the reduction of ensemble members is unnecessary. As part of the overall streamlining of the introduction, we removed the sentence: "To balance the benefits of ensemble forecasting with the constraints of computational resources, we selected ten of the 40 ensemble members used in the operational configuration." (lines 81–82). Instead, we now introduce the setup with: "Further, we explore its [two-moment scheme's] impact within a ten-member ensemble prediction system (EPS), assessing how ensemble-derived metrics can enhance ISSR identification beyond mean-state representation." In the following sections, we refer to the ten-member ensemble only when relevant, without explicitly mentioning the reduction compared to the operational setup. For completeness, this point is briefly addressed in the "Model Setup" section.
- Similar reductions were applied throughout the manuscript and are highlighted as red text changes in the revised version.

**Manuscript changes**

- We shortened lines 178–180 to "The temperature is measured with an accuracy of  $\pm 0.2$  °C and the humidity with an accuracy of  $\pm 3\%$  RH."
- We also removed the more detailed explanation of the humidity measurement technology in the IAGOS case, as it was not essential for the scope of our paper and is covered by referenced sources.

• Comment 2: Since you are introducing an improved version of the two-moment scheme you may spend a little more time on the actual improvement. Would it be worth to shift it from Appendix A to the main text?

**Response**

We agree with the Referee and have moved the relevant parts of Appendix A – detailing the two–moment ice microphysics parameterization – into the main text. This change also aligns with the suggestion from Referee #2. Further details are provided in our response to Referee #2.

• Comment 3: The analysis makes use of ten different "metrics": confusion matrix, FBI, POD, FPR, precision, MCC, ROC, k-out-of-10, Youden index, and F1. You may want to consider reducing the number of metrics and focusing on the most important ones that can be used throughout the paper. If you decide to keep them all, they must be introduced well, either in the methods section (preferred) or in the text when they are first used. Please provide the possible technical ranges of the indices/metrics, common values, and the desired value so that the reader can interpret the metrics.

**Response**

We agree with the Referee that introducing numerous verification metrics solely within the verification analysis section may be overwhelming and could hinder readability. To address this concern, we revised the manuscript in two ways:

- 1) Refocusing the set of metrics: We streamlined the set of evaluation metrics to emphasize the most relevant ones for our application. Specifically, we focus on the Probability of Detection (POD) and the False Positive Rate (FPR), which are central to our analysis, particularly in the context of ensemble evaluation via ROC curves. We removed the Youden Index, as its assumption of equal costs for false negatives and false positives does not generally hold in our case and it was just added as an example. Similarly, we omit the F1 score, which emphasizes performance on the positive condition through the harmonic mean of POD and precision. Since our focus is broader, we will instead retain and discuss precision directly, both for ISSR and non-ISSR cases. The remaining metrics, which we believe offer valuable insight into the performance of our model, are introduced more clearly (see next point).
- 2) Introducing a dedicated Methods Section: We addes a new "Verification Methods" Section to the manuscript, where we define and explain the key concepts and metrics used to evaluate both the deterministic and the probabilistic (ensemble) model. This provides readers with a clearer foundation and reduce redundancy in the verification section (now "Verification Results" Section). To support this, we included a figure (see Fig. 1) that outlines the structure and relationships between the main metrics used. This visual aid has been incorporated into the Verification Methods Section to offer a quick overview, serve as a reference point, and help streamline the text.

In summary, the new section serves two main purposes:

- \* It explains how the spatio—temporal matching between model and observational data was derived, forming the basis for all verification scores.
- \* It introduces the concept of categorical verification, covering both the deterministic model case and the ensemble model case.

We expect these changes to improve the clarity and readability of the manuscript, while maintaining the necessary depth of the analysis.

Figure 1: Overview of categorical verification methods used in this study. (a) Confusion Matrix: Provides a structured summary of how model predictions align with actual observations in a binary classification setting. Each prediction is categorized as a true positive (TP), false positive (FP), false negative (FN), or true negative (TN), depending on its agreement with the observed outcome. This matrix forms the foundation for computing categorical performance metrics such as listed in (b). (b) Categorical metrics: Frequency Bias Index (FBI), Probability of Detection (POD), False Positive Rate (FPR), Precision (Prec), and the Matthews Correlation Coefficient (MCC) offer distinct insights into model behavior as described in Section 4.2. (c) Categorical evaluation of the ensemble prediction system: (i) the discrimination diagram shows two distributions of forecast probabilities; one for the case where the event was observed in the measurements, and one where it was not observed, highlighting the model's ability to separate events by probability; (ii) the Receiver Operating Characteristic (ROC) curve illustrates the trade-off between the POD and FPR across different classification models based on the ensemble's probabilistic event forecast.

**Manuscript changes**

The structure of the new Verification Methods section is as follows:

- \* 4 Verification Methods
  - $\ast\,$  4.1 Spatio–Temporal Matching of Model and Observation Data
  - \* 4.2 Categorical Metrics
  - \* 4.3 Categorical Verification of Probabilistic Model
    - · Discrimination diagram (presented as an unnumbered paragraph and not as a subsubsection to reduce fragmentation)
    - · Receiver Operating Characteristic Curve (ROC) (presented as an unnumbered paragraph and not as a subsubsection to reduce fragmentation)

• Comment 4: The discussion session reads more like a summary. If introducing a dedicated discussion session, then the results of the analysis should discussed by setting them into context with existing literature. For example to discuss differences or similarities in distributions of ISSR and to explain the causes for potential differences.

**Response**

We thank the Referee for the helpful comment and suggestion. As our discussion section currently consists of four subsections, we respond to each individually:

- Subsection 1 "Interrelationships of Results and Application Implications": We agree that this subsection functions more as a summary or conclusion. We have shortened it and merged it with the "Conclusions" section, which should also help make the manuscript more concise.
- **Subsection 2** "Ensemble Verification of ICON 1–Mom EPS": We moved this subsection to the Appendix to streamline the main text.
- Subsection 3 "Model Resolution and Neighborhood Consideration": This subsection aligns well with the classical discussion format suggested by the Referee. We have therefore retained it within a dedicated "Discussion" section. To enrich the discussion, we expanded it by adding a second topic: "Comparing Microphysics-Based and Statistical Approaches to Ice Supersaturation".
- Subsection 4 "Prediction Improvement via Machine Learning": We have relocated this subsection to a newly created "Outlook" section, placed immediately after the "Discussion" section. This new placement provides a more natural fit in terms of content and narrative flow, particularly as it allows us to revisit the topic of neighborhood cell inclusion in a forward-looking context.

**Manuscript Changes**

The adapted dedicated "Discussion" section has been changed as follow:

- A new subsection titled "Comparing Microphysics-Based and Statistical Approaches to Ice Supersaturation" has been added to the revised discussion.
- The paragraph discussing similarities to findings of [2] within the previous subsection "Incorporating the Ensemble Spread" was moved to the "Discussion" section. We titled it "Observed Standard Deviation of RHice"
- All other revisions to the former "Discussion" section are highlighted in red in the revised manuscript.

**Minor comments**

• The manuscript focuses specifically on contrail formation. However, the formation of cirrus clouds and supersaturation is also relevant to the water budget in the upper troposphere and lower stratosphere. You could mention this as an additional motivation for improving the representation of supersaturation.

**Response**

We added two motivating sentences in our introduction, after mentioning that accurately capturing  $RH_{ice}$  is important for contrail modeling and before mentioning the difficulties in  $RH_{ice}$  prediction.

**Manuscript changes**

Beyond contrail modeling, ice-supersaturated regions play a critical role in the development and persistence of cirrus clouds, which are key regulators of the water vapor budget in the upper troposphere and lower stratosphere [6]. Improving the representation of supersaturation is therefore vital not only for contrail modeling but also for capturing the broader impacts of cirrus cloud dynamics on atmospheric moisture and radiative balance [3, 1].

• L40: Could you provide a reference for interested readers?

**Response**

We cite now [9, 4, 8]

**Manuscript changes**

Yet, despite its relevance for climate-relevant processes, RHice remains one of the most uncertain variables in NWP models [9, 4, 8].

• L41: NWP was already introduced in Line 35

**Manuscript changes**

Errors and uncertainties in **NWP** models stem from various factors, ...

• L43: RH\_ice was already introduced in Line 35

**Manuscript changes**

Among these challenges, accurate prediction of  $\mathbf{RH_{ice}}$  remains particularly difficult, ...

• L41-47: These lines contain redundant information by mentioning the uncertainty and lack of in situ observations multiple times. Please revise accordingly.

**Response**

We have removed the redundancies and focus on uncertainties in RHice NWP-forecasts.

**Manuscript changes**

RHice prediction is particularly challenging due to ...

• L52: Maybe "...and returns adjusted values of RH\_ice?" instead of "outputs".

**Manuscript changes**

One way to circumvent these limitations is to develop machine learning methods to derive  $RH_{ice}$  forecast corrections. The resulting correction model receives variables such as temperature,  $RH_{ice}$ , and others, and returns adjusted values of  $RH_{ice}$ .

- L52: "Wang et al. (2025)...." Please note that ERA5 is suspected to be biased in terms of relative humidity, which causes problems in resolving ISSR. This is due to the fact that RH\_ice is clipped to a maximum value, as well as the spatial resolution of the model grid. However, there is no consensus on whether RH\_ice is too low or too high at the tropopause level. Several correction methods for ERA5 have been provided, e.g.,
  - Schumann, U. and Graf, K.: Aviation-induced cirrus and radiation changes at diurnal timescales, J. Geophys. Res.-Atmos., 118, 2404–2421, https://doi.org/10.1002/jgrd.50184, 2013.
  - 2) Schumann, U., Penner, J. E., Chen, Y., Zhou, C., and Graf, K.: Dehydration effects from contrails in a coupled contrail-climate model, Atmos. Chem. Phys., 15, 11179–11199, https://doi.org/10.5194/acp-15-11179-2015, 2015.
  - 3) Teoh, R., Schumann, U., Gryspeerdt, E., Shapiro, M., Molloy, J., Koudis, G., Voigt, C., and Stettler, M. E. J.: Aviation contrail climate effects in the North Atlantic from 2016 to 2021, Atmos. Chem. Phys., 22, 10919–10935, https://doi.org/10.5194/acp-22-10919-2022, 2022a

**Response**

We thank the reviewer for drawing our attention to these interesting studies. We mention one of them in the introduction of the revised manuscript.

**Manuscript changes**

Previous studies have also examined corrections to ERA5 reanalysis  $RH_{ice}$ , particularly in the context of estimating the climate effects of aviation contrails (e.g., [13])

• L54: At the end of the sentence: "...when validated against test data"?

**Manuscript changes**

..., showing RHice mean absolute error improvements when validated against test data.

• L84: NWP already introduced

**Response**

The corresponding paragraph was deleted due to other comments.

• L92: Throughout the manuscript, there are multiple versions of RH\_ice. There is: RHice and RHice. Figures use RHi. All of these combinations are sometimes in italics and sometimes not. Choose one version and stick to it throughout the script.

**Response**

We stick to RHice now in the entire manuscript.

• L96: What are the typical challenges? They are known to you, but since you are making this point, please briefly mention them. Or are they given in the next sentences? If so, I would suggest to write, ... with forecast applications, such as data assimilation and model uncertainties, the interpretation of the resulting forecast..."

**Response**

In response to the comments from Referee #2, we have removed the paragraph containing general statements on ensemble forecasting.

• L126: abbreviate "relative humidity of ice"?

**Response**

Implicitly solved due to changes in the Model Section.

• L140: and elsewhere. To facilitate understanding of the paper, clearly define your different model setups and provide unique abbreviations for each configuration. ICON (for the operational 1-Moment scheme), ICON 2-Mom (for the new 2-Moment scheme), and ICON EPS 2-Mom. Then, stick to these abbreviations. The current version uses various abbreviations and paraphrases. Sometimes, the 2-Mom model is also called the deterministic model. These variations make it difficult to understand the paper's content and cause unnecessary ambiguities.

**Response**

Until the Model Setup subsection, we consistently describe the model specifications in full. At the end of that subsection, we introduce abbreviations for the different models evaluated throughout the study. To enhance consistency and readability, we have removed italic formatting from these abbreviations. Throughout the manuscript, we adhere to these abbreviations to facilitate clarity and ease of reading.

**Manuscript changes**

The model outlined forms the basis for the evaluations performed in this study and will be referred to as ICON 2–Mom EPS in the remainder of this study. Since the dedicated ICON forecasting system does not consist of an additional deterministic model run, we use individual members of the ensemble as approximates to a deterministic model setup for our evaluation, denoted by ICON 2–Mom in the following. Similarly, the operational ICON with the one–moment ice microphysics scheme is denoted by ICON 1–Mom.

• L164 and elsewhere: For ranges, e.g., 8.5-12.5, use spaced en-dashes "-" see ACP style guide

**Response**

We have revised the entire manuscript accordingly.

• 169 - 170: Why is ICON 2-Mom now cursive? Please see comment above.

**Response**

We have revised the entire manuscript accordingly.

• L184: What does "TEMP BUFR" mean? Enter the full name here. Explaining BUFR in the next sentence is insufficient and too late.

**Manuscript changes**

Radiosonde observations are typically conducted twice daily, with balloon ascents around 0 UTC and 12 UTC. The resulting data are stored in standardized binary files known as Binary Universal Form for the Representation of meteorological data (BUFR), a format developed by the World Meteorological Organization (WMO) to encode and transmit various types of weather observations. These files contain TEMP reports, which include a structured set of atmospheric measurements such as temperature, pressure, humidity, and wind speed and direction at multiple vertical levels. TEMP BUFR files serve as the standardized source of radiosonde data used in this study.

• L185: It is usually first the long name, followed by the abbreviation in brackets.

**Manuscript changes**

The resulting data are stored in standardized binary files known as Binary Universal Form for the Representation of meteorological data (BUFR)

• Fig2: The subpanels and labels are small but still legible. Would it not make sense to split the plot and place the individual plots in the positions where they are discussed? This would also prevent flipping back and forth through multiple pages when referencing back from Section 4. The authors may want to consider this.

**Response**

We thank the Referee for this suggestion. While we understand the concern, we prefer to retain the current layout, as it visually emphasizes the relationship between the subplots. We also expect that the final double-column format will improve readability and reduce the need for page flipping.

• L209: Please explain which model you mean with "dedicated model". ICON 1-Mom, 2-Mom?

**Response**

We have removed the term "dedicated" and now explicitly refer to the two-moment ice microphysics scheme. Additionally, we have contextualized the subsequent steps by consistently using the model abbreviations introduced earlier in the Model Setup subsection.

**Manuscript changes**

We evaluate the  $RH_{ice}$  predictions of ICON equipped with the new two-moment ice microphysics scheme in two steps. First, we verify the deterministic model, ICON 2–Mom, which includes a comparison with ICON 1–Mom. Second, we evaluate the ensemble prediction system, ICON 2–Mom EPS.

• L213: Call it ICON 2-mom?

**Manuscript changes**

Verification of Deterministic Model ICON 2-Mom

• L215: Why ICON 1-mom now in italics?

**Response**

Originally, the abbreviation was formatted in italics at this point in the text to indicate its introduction. However, we have revised our formatting approach and no longer use italicized abbreviations anywhere in the manuscript.

• L224: Is the new paragraph required?

**Response**

We have removed the unnecessary line break to improve the flow and formatting of the section.

• L224: Is "density tail" the correct term? I have not found it in the literature. Would you call it the "tail of the density distribution"?

**Manuscript changes**

Pronounced differences emerge in the tail of the density distribution,  $\dots$

• L224: "operational system"? you mean ICON 1-mom?

**Response**

We clarify this by consistently using the model abbreviations introduced earlier.

**Manuscript changes**

ICON 1-Mom exhibits a sharp peak near 100 %, ...

• L230: "The ICON grid employed features a horizon..." Please check the grammar of this sentence.

**Manuscript changes**

The ICON grid used in our model setup has a horizontal resolution of approximately 26 km and a vertical resolution of 200–300 m within the altitude range of 8500–12500 gpm.

• L232-233: Why do you use the closest match to the radiosonde station instead of the actual position of the radiosonde? Radiosondes drift horizontally by several kilometers and may end up in a different grid cell.

**Response**

We thank the Referee for this insightful comment. It is indeed true that radiosondes can drift horizontally by several kilometers during ascent. However, given the horizontal grid resolution of 26 km, they only occasionally cross into a different grid cell. Based on discussions with our colleagues in data assimilation, such drift has a negligible impact on the verification scores. While it is technically possible to account for radiosonde drift, doing so would require additional effort due to the triangular structure of the ICON grid. Considering the minimal expected effect, we chose not to implement this correction. As a result, our verification scores can be interpreted as conservative estimates, which may slightly improve if drift were explicitly considered.

• Sec 4.1.1 and 4.1.2: What are the fundamental differences between these subsections? Would it not be better to start with 4.1.2 and explain how the data is extracted, and then analyze and compare the data? Additionally, both could be done in one subsection ending in a less fragmented text.

**Response**

We appreciate the Referee's suggestion and have incorporated it into the overall restructuring of the manuscript. The first part of Section 4.1.2 –describing the spatio-temporal matching of model and observational data – has been relocated to the newly introduced "Verification Methods" section. The description and analysis of the scatter plot (now more correctly termed 2D histogram) remain in a dedicated subsection within the "Verification Results" section. While we acknowledge that this structure may appear somewhat fragmented, we chose to retain it in order to highlight the distinct types of verification approaches used in our study.

• L241: "An ICON spin up time of a minimum of 6 hours was required" This does not fit here. Would it be better placed in the introduction of the ICON model?

**Response**

Valid point. In the "Model Setup" section, we already state: "It starts from the operational analysis, which is based on the one-moment ice microphysics scheme, so that we require a spin-up time of at least 6 hours in our evaluations below to build up ice supersaturation." We have therefore removed the redundant and misplaced sentence at line 241, as suggested by the Referee.

To improve clarity, we added unnumbered paragraphs in the "Spatio-Temporal Matching of Model and Observation Data" section to distinguish between the radiosonde and IAGOS cases. Each paragraph now includes the following clarification regarding temporal matching:

**Manuscript changes**

Radiosonde case: For temporal matching, the start time of the accent was used as a reference, and we select the corresponding ICON simulation whose initial time is closest to the observation time minus the required lead time. Since the simulation provides hourly forecasts, this approach ensures temporal matching to the nearest hour. The exact lead time is explicitly stated in all evaluations and never below the required spin—up time of 6 hours.

**IAGOS case:** For temporal matching, the minimum lead time was fixed at 6 hours to account for the required ICON spin-up. Since flights span several hours, different ICON simulations were used, each selected based on the initial time closest to the observation time minus the 6-hour lead time. As ICON simulations are initialized in 6-hour intervals, this approach may result in a maximum temporal mismatch of  $\pm$  3 hours.

• L247: "...simple scatter plot." Please explicitly mention which plot you are referring to so that the reader can more easily identify it. In my opinion, showing it at the end of the next sentence is too late. I found myself wondering where to look.

**Manuscript changes**

We examined the 2D histograms of RHice of spatio-temporally matched points between Vaisala RS41 radiosonde data and ICON forecasts (Fig.1(c)).

• L262: Why is the first part in brackets and the rest in the subscript?

**Response**

We have removed both the subscript and the brackets around the event labels, and now state the thresholds clearly in the main text.

**Manuscript changes**

In the remainder of this study, we consider events of the type

 $RH_{ice} > threshold,$

with threshold  $\in \{100\%, 105\%, 110\%, 120\%\}$ .

• L265: You may state the possible range of FBI and that a value close to 1 would be desirable.

**Response**

We included this in the "Verification Methods" section, and in the corresponding figure (Fig. 3).

• L277: "...the POD increases from about 0.4 for ICON 1-Mom..." I guess this is for the 100% threshold? You should mention that.

**Manuscript changes**

For ISSR events (RH $_{\rm ice} > 100$  %), the POD increases from approximately 0.4 for ICON 1–Mom to around 0.6 for ICON 2–Mom,

• L282: "..., also known as sensitivity,..." Does this refer to the probability of detection? If so, please introduce the term "sensitivity" when defining the POD.

**Manuscript changes**

The probability of detection (POD, also known as sensitivity) ...

• L232: "specificity" is not defined. Do you mean sensitivity?

**Manuscript changes**

The false positive rate (FPR, also defined as 1-specificity) ...

• L301: "Another way..." if I'm not mistaken, no method has been mentioned so far to address the imbalance. At least, none has been mentioned explicitly. Please check and revise the script accordingly.

**Response**

We thank the Referee for pointing this out. We have revised the manuscript accordingly: the Matthews Correlation Coefficient (MCC), which accounts for all four components of the confusion matrix and is well-suited for imbalanced datasets, is now introduced in the "Verification Methods" section and referenced again in the "Verification Results" section.

**Manuscript Changes**

- Verification Methods section: The Matthews correlation coefficient (MCC) is a composite measure that accounts for all four components of the confusion matrix simultaneously.
- Verification Results section: The MCC shown in Fig. 4(e) summarizes overall classification performance.
- L306 here and elsewhere: minus signs with -.

**Response**

We have revised the entire manuscript to consistently use the correct minus sign notation (-) as suggested.

• L321-322: You may also mention that humidity varies greatly in the atmosphere, and radiosondes have a much higher spatial and temporal resolution than models.

**Response**

Thank you for this insightful comment. We address this point in more detail in our responses to major comments 5 and 8 from Referee #2, where we discuss the implications of humidity variability and the resolution differences between radiosonde observations and the model.

• L324-327: What do you mean by "post-processing"? Please specify. As I understand it, you are proposing a new two-moment scheme that allows for supersaturation. Your previous analysis showed that the two-moment scheme performs better than the one-moment scheme. Does the new two-moment scheme just need more refinement or adjustment instead of additional post-processing steps? Alternatively, please explain what is meant by "postprocessing."

**Response**

That's a correct understanding of the analysis so far. We understand that the sentence might be a stumbling block for readers following the manuscript closely.

The sentence is intended to provide a transition towards the analysis as it occurs in the following sections, where a single decision on ISSR classification is formed by incorporating all ensemble member values. Refining the two-moment scheme is not straightforward because its parameterization reflects physical realities where small tweaks to balance this histogram might do more harm than good. Therefore, the ensemble comes in a second step into play.

**Manuscript Changes**

Thus, we further analyze the ensemble's ability to classify ISSR and non-ISSR conditions below.

• L329-330: "distinguish between ISSR and non-ISSR conditions (or higher supersaturation)," Should it be "non-ISSR and ISSR (or higher supersaturation) conditions"?

**Response**

In the revised manuscript, we have clarified the phrasing to avoid ambiguity. It now reads: "Figure 5(a) shows the conditional distributions of forecast probabilities for observed and non-observed events (events are defined as  $\mathrm{RH_{ice}} > 100\%$  and higher thresholds)."

• L330-332: Why is non-ISSR not an option? As I understand it, the critical point is non-ISSR versus ISSR. At some point, the degree of supersaturation might no longer be relevant in determining whether a contrail can form, but rather, the duration that the ISSR and a potential contrail persist.

**Response**

Regarding the first point: non-ISSR is indeed considered in the discrimination diagram, which evaluates forecast probabilities for both *observed* events (ISSR) and non-events (non-ISSR). This diagram is a graphical tool used to assess the discrimination ability of a probabilistic model with respect to a binary classification task, and by design, the framework inherently includes non-ISSR as the complement of ISSR.

Regarding the second point: we agree that the persistence of contrails depends on the duration of ISSR. However, there is evidence that the degree of ice supersaturation also plays a significant role in contrail persistence. Therefore, we include the detection of higher ice supersaturation events in our analysis.

We now clarify this when first defining the events considered in this study:

**Manuscript changes**

In addition to the duration of ISSRs, pronounced ice supersaturation has been associated with the persistence of contrails [13]. While this link is relatively weak, relative humidity remains the dominant factor in contrail-cirrus evolution, governing both the total ice mass and total extinction [15]. Given its relevance, this study focuses on ice supersaturation events (RHice > 100%) and on cases of pronounced supersaturation (RHice  $\gg$  100%).

• Fig4 (b) has no title. Since you provided one for the inserted plot, you should also provide one for the main graph.

**Response**

Thank you for this attentive comment. As a general rule, we prefer to title figures within their captions rather than directly on the plots. However, as a graphical compromise, we have now added the label "Deterministic models" within the inset plot of Fig. 5(b) to improve clarity.

• L345: Would it be appropriate to start the following paragraphs with a new heading, such as: "Categorical Scores of ICON 2-Mom ENS"? You mentioned investigating the continuous

values first and then switching to the metrics. This would also keep it consistent with section 4.1. It's just a suggestion.

**Response**

Thank you for the thoughtful suggestion. We agree that introducing a new subsection improves clarity and enhances consistency with the structure of the original Section 4.1 (now Section 5.1). It also aligns well with the organization of the newly added "Verification Methods" section (Section 4.3). Accordingly, we have restructured this part of the manuscript and introduced dedicated subsections.

**Manuscript Changes**

New subsections with headers:

- **5.2.1** ISSR/non-ISSR Discrimination Ability
- **5.2.2** Threshold-Dependent Performance
- Figure 5 goes over half of the page. You may want to reduce its size and incorporate some of the explanations into the text.

**Response**

In alignment with our metrics reduction, we removed Fig. 5(b) along with the  $F_1$  score. As a result, the remaining subplots were adjusted to quadratic format, which slightly increased the figure size. However, since the caption was shortened and one subplot removed, the overall space occupied by the figure and its caption has been reduced.

• Fig 7: Please explain "ROC stratification". This has not been done in the text.

**Response**

We thank the Referee for pointing this out. To improve clarity, we have removed the term "stratification" and instead explicitly described how the ROC curves are grouped.

**Manuscript Changes**

ROC curves on sample subsets grouped and color-coded by their standard deviation (std) values.

• L444: Earlier, you used a space "" between the number and the percent sign. From the ACP Submission Guidelines: Spaces must be included between number and unit (e.g. 1 %, 1 m). Please revise the manuscript accordingly.

**Response**

We have revised the entire manuscript to ensure consistent use of spaces between numbers and units (e.g., 1 %, 1 m), in accordance with the ACP Submission Guidelines.

• L452-456: "When comparing [...] In conclusion, even when the model exhibits high confidence, as reflected by a low standard deviation, the histogram still displays intermediate supersaturation. This suggests that certain ISSRs can be well predicted." I have a difficult time following this line of reasoning. Please explain better and potentially rewrite. Do you mean that the peak around RH\_ice = 100 % is not fully resolved and only closely resembles the radiosonde observations? Please define what you mean by "certain." Does this refer to cases with high, medium/intermediate, or low supersaturation? Which ISSR would be missed and which are well represented?

**Response**

By certain ISSRs, we refer primarily to those occurring near thermodynamic equilibrium (RH $_{\rm ice} \approx 100$  %), typically associated with mature cirrus clouds and weak supersaturation. These are well represented due to their stable microphysical behavior.

In contrast, "young or short-lived cirrus clouds ... often form in regions of high ice supersaturation, driven by upward motion from gravity waves or deep convection. These young clouds experience rapid crystal growth due to significant mesoscale temperature fluctuations caused by gravity waves, which create high spatio—temporal variability in supersaturation. The fluctuating vertical motions and ice crystal concentrations make forecasting cloud evolution difficult. As a result, young and short—lived cirrus clouds introduce significant uncertainty in predicting supersaturation, as the microphysical processes are highly dynamic and rapidly changing."

These points are discussed at the end of the corresponding subsection in the original manuscript.

**Manuscript changes**

To improve clarity and strengthen the link to the statistical observation that certain ISSRs are well predictable, we have reordered the paragraphs so that the discussion of which ISSRs are well predictable now follows directly afterwards.

• L461: What is the ROC stratification approach? Stratification did not appear before, only in the caption of Fig. 7, which is insufficient.

**Manuscript changes**

Our approach to grouping ROC curves by the ensemble spread of  $RH_{ice}/100\%$  does not currently incorporate temperature, but may do so in future studies.

• L467: I would avoid using the abbreviation "MTFs." It is only used one more time.

**Response**

We have removed the abbreviation "MTFs" and now refer to the term in full.

• L483: Again, it's just one paragraph in a new subsection. Couldn't it be discussed together with the radiosonde observations? A more important question: Do you need the comparison

with the IAGOS data at all? What additional information does 4.2.4 provide compared to the radio-soundings? If you want to take advantage of the coverage over the Oceans and at flight levels, where most of the commercial aircraft are operating, then you should spend more time on this analysis and explain this. But subsection 4.2.4 is very brief and does not provide new conclusions

**Response**

We thank the Referee for this valuable comment. The IAGOS data provide an independent valuable source of  $\mathrm{RH}_{\mathrm{ice}}$  observations, based on a distinct horizontal sampling strategy compared to radiosondes. While the analysis does not yield additional conclusions beyond those derived from radiosonde verification, it serves to corroborate our findings using a complementary dataset.

Including IAGOS data is particularly useful given that ICON tuning was performed with reference to radiosonde observations. This cross-validation across independent observational platforms strengthens the robustness of our verification results.

We have added a sentence to the paragraph to emphasize that this cross-validation strengthens the robustness of our results across different observational platforms.

**Manuscript changes**

These findings strengthen our verification insights across different, independent observation systems.

• L488: "...radiosonde data (Fig.8)" may suggest that the plot contains radiosonde data, but none is plotted. Perhaps rephrase. Alternatively, add the radiosonde data to the plot. This may make a visual comparison easier.

**Response**

Thank you for pointing this out. We have revised the sentence to clearly state that radiosonde data are not included in Fig. 8. The discussion now focuses solely on the comparison between IAGOS and ICON density profiles.

Additionally, we have clarified the text and figure references regarding the ROC curve comparison to avoid any ambiguity.

**Manuscript changes**

Nevertheless, up to  $\mathrm{RH_{ice}} > 120$  %, the shape of the ROC curves (see Fig. 8) derived from the IAGOS data closely resembles those derived from the radiosonde data (compare Fig. 5(b)).

• L662: units not in italics

**Response**

We have corrected the formatting at L662 in the original manuscript and ensured that units are consistently set in upright (non-italic) font throughout the manuscript.

• In the entire reference section the DOIs are missing.

**Response**

We have added the missing DOIs throughout the entire reference section.

**Reviewer 2**

**General Comments**

The study presents an improved Two-Moment microphysical scheme and then evaluates the performance of the ICON model equipped with this scheme in predicting ice supersaturation up to 48 hours in advance. The evaluation is carried out using observation data measured by radiosonde and aircraft as references. The performance of the model with this new parameterization is also compared to that of the operational One-Moment version of the scheme.

The study is part of efforts to improve flight routing in order to avoid areas of ice supersaturation. Assessment scores that are particularly relevant and well suited to this objective were used. The results show that the Two-Moment scheme provides better performance than One-Moment one for ISSR forecasts. The authors also explored a machine learning approach, which proved promising. This work is of high quality and importance, and it represents a significant contribution to ISSR forecasting. The paper fits the scope of ACP. I highly recommend its publication, but some improvements are necessary.

Overall, the manuscript could benefit from a more concise presentation. Some statements are repeated throughout the text, and certain elements currently included in the Introduction would be more appropriately placed in the Model setup Section. In addition, a significant issue remains concerning the evaluation of the model against fine-scale observations used as a reference, as well as the treatment of the uncertainty associated with these observations.

**Response**

We sincerely appreciate the Referee's constructive and thoughtful feedback. In line with the suggestions regarding presentation, we have revised the entire manuscript to improve conciseness and eliminate redundancies. The "Introduction" section was also restructured, with specific adjustments made in response to Major Comment 1. Please refer to our detailed reply there for further information.

The two main content-related concerns are in detail addressed in the responses to the following specific major comments:

- Evaluation of the model against fine-scale observations: See responses to Major Comments 5 and 8.
- Treatment of the uncertainty associated with these observations: See response to Major Comment 7.

The Referee's concerns are about the representativeness error and the measurement error (instrument noise), the sum of which is often called the observation error. In most cases, the representativeness error dominates, especially when comparing fine-scale observations to coarser model outputs.

We found the Referee's comments helpful in refining and correcting some of our explanations in certain sections of the manuscript. In particular, we aimed to more clearly differentiate between representativeness errors and model errors. These clarifications did not affect the validity of the results themselves.

**Specific Major Comments**

- Comment 1: The introduction could be presented more concisely:
  - a. Lines 69 to 100 should be limited to presenting the ensemble forecasting system as described, and include a bit more details on the one-moment scheme, mentioning its limitations, and then explaining the motivation for transitioning to a two-mode scheme.
  - b. The sentence spanning lines 69–77 should be moved to the end of the Introduction; otherwise, the aims of the work are introduced too early.
  - c. The paragraph from lines 80 to 82 should be moved to the Model Setup section.
  - d. The paragraph from lines 101 to 104 should be merged with the last paragraph of the introduction, for example as follows: "This work consists of presenting a new version of the scheme ... and assessing its performance against observations and the old version. It is structured as follows: ...".

**Point-by-point Response**

We thank the Referee for their constructive comments on improving our introduction. We list our adaptations accordingly:

- a. We removed the general statements about ensemble forecasting (i.e., lines 83–88) and added more detail on the one-moment scheme, including its limitations and the motivation for transitioning to a two-moment scheme.
- b. We moved the sentence spanning lines 69–77 to the end of the introduction, so that the aims of the study are introduced after the necessary context has been established.
- c. We removed lines 80–82 from the introduction and incorporated the content into the "Model Setup" section.
- d. We merged the paragraph from lines 101–104 with the second last paragraph of the introduction, following the Referee's suggestion to streamline the structure and clarify the study's objectives.

**Manuscript Changes**

The second half of the revised introduction follows a streamlined structure:

- We introduce the motivation for employing a two-moment ice microphysics scheme in ICON, emphasizing its advantages over the operational one-moment approach.
- We briefly justify the use of an ensemble prediction system, focusing on its specific relevance to ISSR prediction rather than general ensemble forecasting benefits.
- We now present the study's objectives more clearly and at a later point in the introduction, referring to both the two-moment microphysics scheme and the ensemble framework. This section has been shortened and streamlined, occupying approximately 20% of the introduction to ensure a concise and proportionate focus on our specific contribution within the broader context.

• Comment 2: Description of the parameterization: The appendix (lines 609–610) indicates that this parameterization is a simplified version of that of Köhler and Seifert (2015), but it does not specify how it differs, and this is also not clarified in Section 2.1. I suggest summarizing Sections A1, A2, and A3 – within the main body of the article in Section 2.1, to improve readability, understanding, and reproducibility, especially since this parameterization does not appear to have been published elsewhere. This could be achieved without significantly lengthening the manuscript.

**Response**

We agree with the Referee that the understanding of the new parameterization is increased when moving some details to the main text, so we moved Appendix Sections A1, A2, and A3 to the "Model" section. This is also in accordance with Referee #1.

The main differences to Köhler and Seifert (2015) are the use of only one ice particle mode (mentioned in the manuscript "The two–mode representation in KS15 is omitted for computational efficiency, as are the timestep refinements for homogeneous nucleation.") and the use of the heterogeneous nucleation parameterization of Ullrich et al. (2017) [14].

**Manuscript changes**

The "Model" subsection "Two–Moment Cloud Ice Microphysics Parameterization in ICON" was augmented accordingly.

• Comment 3: The presentation of the evaluation metrics is currently scattered throughout the Results section, which somewhat affects the readability. It may be clearer and more convenient for the reader if the authors dedicate a specific section to these metrics and concisely.

**Response**

In alignment with Referee #1's suggestion, we have introduced a dedicated "Verification Methods" section. The original section titled "Verification Analysis" has been renamed to "Verification Results". The new methods section presents the main categorical evaluation metrics used in our study along with their relationships. To support a concise and accessible overview, we have added a figure (Fig. 1 in this document, Fig. 3 in the revised manuscript) illustrating the key concepts and their connections.

**Manuscript changes**

- 4 Verification Methods
  - \* 4.1 Spatio-Temporal Matching of Model and Observation Data
  - \* 4.2 Categorical Metrics
  - \* 4.2 Categorical Verification of Probabilistic Model
    - · Discrimination diagram (presented as an unnumbered paragraph instead of a subsubsection to avoid fragmenting the relatively short subsection too much)
    - · Receiver Operating Characteristic Curve (ROC) (presented as an unnumbered paragraph instead of a subsubsection to avoid fragmenting the relatively short subsection too much)

• Comment 4: Line 154: You mention that "the system includes stochastic perturbations of selected parameterisations". Could you specify which parameterisations are perturbed in your simulations? Furthermore, given that the model may become unrealistic with certain parameter values, could you clarify how the stochastic perturbations are applied? Is it limited to an area of the space of the parameter values, relevant to ISSR forecasting, in order to ensure that the model produces realistic values for these forecasts?

**Response**

That's an excellent point. We have expanded the manuscript to specify which parameterizations are stochastically perturbed, and added the reference which describes parameter ranges and more details.

**Manuscript changes**

In addition to initial condition perturbations, the system includes stochastic perturbations of selected physical parameterizations which are known to be sensitive. Thereby, different components of the system are perturbed, including gravity waves, convection, microphysics, the cloud scheme, turbulence and land surface. For example for convection, well-know parameters such as the entrainment rate or the excess of moisture or temperature used in the ascent of a test parcel are targeted. For the global ensemble system, these physical parameters are randomly perturbed for each ensemble member with time-dependent perturbations varying sinusoidally within their range. The randomisation is accomplished by a phase shift of the sinusoidal wave depending on the ensemble member ID (for more details see Chapter 13.2 in Reinert et al., 2025). This approach introduces variability among ensemble members while preserving the consistency of individual forecast trajectories. The combined perturbation strategy ensures a realistic representation of forecast uncertainty, which is crucial for assessing the sensitivity of contrail formation potential to meteorological variability.

As a third source of uncertainty, the sea-surface temperatures over oceans are perturbed in the initial conditions.

- Comment 5: Section 4.1.2 from line 230 to line 235: You mentioned that, in order to perform the spatio-temporal comparison, the observations (radiosonde and IAGOS data) were made comparable to the model grid by vertically interpolating the observations to the model levels. Unless I am mistaken, no horizontal spatialization was performed, only colocation. Consequently, these vertically interpolated observation values can be considered as relatively local.
  - a. Could you discuss the relevance of their approach of comparing these vertically interpolated values with those of the model, which has a horizontal resolution of 26 km?
  - b. Communities working on precipitation generally address resolution differences using spatial kriging, but this requires closer stations in order to calculate spatial covariance. I imagine that this type of approach is not usable in your study because measurements are not made everywhere at the same time and the distance between measurement points is very large. Consequently, it is important to discuss the implications of the differences in resolution on the results.

**Response**

Thanks for the additional perspective. ISSRs tend to stretch horizontally far more than vertically, with typical horizontal scales of around 140 km as seen in [12], however with large standard deviation of 250 km [5], whereas vertical scales are between 200 m to 500 m. Due to this difference by orders of magnitudes we felt that our approach of tackling the horizontal and vertical axes separately was justified. Kriging or similar methods are well justified and should work well if we use the 26-km-spaced ICON grid as our point of reference, but, as pointed out by the Referee, going in the opposite direction and interpolating between observations is not feasible due to the sparse data situation. Somewhat embarrassingly, even a simpler interpolation approach on the ICON grid introduced a whole slew of code issues to the extent that we weren't 100% confident in the results anymore. We expect that interpolation would improve the results further, albeit by an indiscernible amount.

Although ISSRs represent a multiscale phenomenon involving processes from synoptic to turbulent scales, they can still be roughly partitioned into two classes. Large, stable ISSRs are associated with slow, steady ascent and near-equilibrium conditions, while smaller, short-lived ISSRs are driven by dynamic processes such as gravity waves or convection. Stable ISSRs are well-represented by our method of matching, and thus contribute positively to the verification scores. Meanwhile small-scale ISSRs are difficult to capture regardless of how matching was performed due to the coarseness of the ICON grid.

We realize that the choice of how model and observations are matched does have a small impact on the overall validation scores, but are confident that our choice forms a rather conservative baseline.

**Manuscript changes**

We added some more discussion and clarifications to the revised manuscript:

- In subsection "Spatio-Temporal Matching of Model and Observation Data, Radiosonde Data": No horizontal interpolation was applied. However, the impact is expected to be minimal, as typical horizontal scales of ISSRs are on the order of 140 km [12].
- When discussing the rank histogram of ICON 2-Mom EPS: Fig. 2(c) shows the resulting histogram for the subset of samples where the observed RHice is above 50 %. We consider this restricted rank histogram because ICON tends to underestimate very low humidity values, which are not the subject of this study but would obscure the relevant behavior (also reflected by the RHice histogram in Fig. 1(b, bottom)). The histogram exhibits a U-shape, indicating underdispersion, i.e., the ensemble fails to capture the full variability present in the observations. This behavior is partly due to spatial averaging over model grid cells, which tends to smooth out extremes. However, counteracting this, so-called upscaling effects of the model tend to display small-scale physical behavior on the model scale. Thus, insufficient parameter perturbations may be another reason, together with the lack of subgrid-scale gravity waves and the use of climatologically prescribed aerosol fields, both of which constrain variability in ice nucleation conditions.
- At the end of subsection "Model Resolution and Neighborhood Consideration": We expect that using a finer grid for ICON predictions may enable such an approach, and most likely improve the overall verification scores.

• Comment 6: Ligne 258: In connection with the previous comment, is it appropriate to systematically use the threshold 100 % to define ice supersaturation in the model? Given the model's horizontal resolution (26 km), local areas of supersaturation may exist even when the grid-cell average remains below the saturation threshold. Have you tested the impact of using a slightly lower threshold, for example between 90 % and 100 %, to define ISSRs in the model, on its performance?

**Response**

We acknowledge that subgrid-scale variability may lead to localized ice supersaturation even when the grid-cell mean remains below 100~% RHice. However, the two-moment microphysics scheme implemented in ICON explicitly prognoses specific ice mass and ice particle number density, allowing phase relaxation time and supersaturation to emerge from physically consistent processes. Applying a threshold below 100~% would risk misclassifying grid-scale conditions and artificially inflating ISSR occurrence. Given the model resolution, uncertainties exist in both directions, and lowering the threshold could lead to an overestimation of ISSRs.

This is particularly evident in the  $RH_{ice}$  histogram of the two-moment scheme shown in Fig. 1(a, bottom) in the revised manuscript, where a pronounced local maximum appears tightly around  $RH_{ice} = 100$  %, closely matching the observed distribution.

For these reasons, we retain the 100 % threshold to ensure physical consistency with the microphysics scheme and avoid introducing bias through arbitrary threshold adjustments.

Also interesting: The CatBoost model has learned to predict ISSRs based on the  $\rm RH_{ice}$  values of the 10 ensemble members. If a lower threshold had provided a predictive advantage, the model would likely have captured this. However, its predictions closely resemble those of the k-out-of-10 models, where the ISSR threshold applied to individual members matches that used for the observations. This suggests that lowering the threshold does not lead to improved performance.

• Comment 7: In the description of the observations, an uncertainty is mentioned, but it is not discussed further. Could the authors consider taking this uncertainty into account in the evaluation, as they did in Section 4.2.3 for the model by incorporating its ensemble spread? In other words, have you examined whether adjusting the threshold according to observations' uncertainties range, could impact the model performance score?

**Response**

We appreciate this suggestion, however, the BUFR data itself does not contain quantitative information about actual measurement errors or uncertainty for relevant physical quantities of interest, and radiosonde data makes the bulk of the data. The 3% uncertainty estimate from [16] would be the only thing we could incorporate, but given that this refers to a normal distribution around the actual value, and we're only interested in binary classification, only values around the threshold are of interest, and in this band around the threshold the sample uncertainties are almost identical. Given this data situation, we are even uncertain about in which direction the threshold ought to be adjusted without introducing a bias to the verification.

This differs from the ensemble spread situation where we take a look at the model output alone and derive meaningful information, and also from regression problems where the uncertainty could be easily included.

Additionally, we use only Vaisala RS41 radiosonde observations, as they were found to perform best in the large-scale intercomparison campaign [16].

Overall the impact of the measurement uncertainty is minuscule compared to representative errors or model forecast errors.

Nevertheless, we agree with the Referee that in general it would be nice to receive and include the specific uncertainty of each observation in the verification. Therefore, more elaborated verification measures should be developed, similar to the principles of data assimilation, where the uncertainty of different observations is estimated and included in the assimilation procedure. In the context of verification, this could look like weighting the spatio—temporal matching points of model and observation within a verification score according to the uncertainty range of each observation.

However, this goes beyond the scope of this study and is also usually not performed in standard NWP verification, mainly because the resulting error is orders of magnitude lower than errors arising from representativity issues (occurring due to different resolutions and sparse data), and model errors.

• Comment 8: In lines 324-325: the authors state: "We also observe a more pronounced negative bias within the rank histogram, indicating that the model tends to underestimate RHice more often than it overestimates RHice." Could this observation not be explained, at least in part, by the difference in spatial resolution between the observations and the model?

**Response**

The difference in resolution most likely is one reason for the general U-shape of the rank histogram, but not specifically for the more pronounced negative bias. We believe that resolution issues alone should result in a symmetric U-shape, reflecting random undersampling of subgrid-scale variability in both directions, i.e., under- and overestimation.

The pronounced negative bias in the rank histograms (Fig. 2(c) and 2(f) in the original manuscript) is mostly caused by the model behavior at very low humidity values, compare Fig. 2 below. When considering the rank histogram for different regimes of observed  $RH_{ice}$ , the negative bias is primarily given in the regime of low humidity ( $RH_{ice} < 10\%$ ) and also in the regime of ice supersaturation, while for the bulk of intermediate humidity values the rank histogram is slightly U-shaped and even slightly left-biased.

However, from biases in rank histograms we can not infer magnitudes.

The continuous  $RH_{ice}$ -histograms seen in Figure 1(b, bottom) in the main manuscript show that in actual units (i.e. non-ranked, but also not spatio-temporally matched) both model and observed  $RH_{ice}$  values follow a similar density shape and have similar mean values.

Motivated by the comparison of the  $RH_{ice}$ -histograms in Figure 1(b, bottom), we replaced the rank histograms in the manuscript to only include samples where observed  $RH_{ice}$  values are greater than 50 %. We feel that this step is justified given our focus on prediction of ISSRs, whereas very low  $RH_{ice}$  regions dominate the overall number of samples (and thus the shape of the rank histogram) but are not too interesting from a modeling perspective. In the regime  $RH_{ice} > 50\%$ , the negative bias in the rank histogram is much smaller than in the full humidity regime. We suggest that the remaining bias is primarily due to model physics limitations, such as parameterizations and the use of climatologically prescribed aerosol fields.

Thus, given these more divers perspectives and the results from Figure 1(b, bottom) in the main manuscript, spatial resolution effects may not be the (main) underlying cause.

In particular, during the development and tuning of the ice microphysics scheme, so-called upscaling effects arise – where small-scale physical processes are statistically represented at the model grid scale. These effects can imprint realistic small-scale behavior onto the coarser model resolution, helping mitigate resolution-induced biases.

We adapted and clarified the discussion of the rank histograms in the revised manuscript to reflect this multifactorial interpretation. Furthermore, in Fig. 2(c) and 2(f), we now show the rank histograms restricted to  $RH_{ice} > 50\%$ , to focus on the most relevant  $RH_{ice}$  regime and on the most relevant model behavior.

Figure 2: Rank histogram grouped according to three different observed  $RH_{ice}$  regimes, in all cases compared to the (same) full rank histogram in gray: spatio-temporal matched samples with (a) observed  $RH_{ice} < 10\%$ ; (b) observed  $RH_{ice} \in [10\%, 100\%]$ ; (c) observed  $RH_{ice} > 100\%$ .

**Manuscript changes**

Fig. 2(c) shows the resulting histogram for the subset of samples where the observed  $RH_{ice}$  is above 50 %. We consider this restricted rank histogram because ICON tends to underestimate very low humidity values, which are not the subject of this study but would obscure the relevant behavior (also reflected by the  $RH_{ice}$  histogram in Fig. Fid. 1(b, bottom)). The histogram exhibits a U-shape, indicating underdispersion, i.e., the ensemble fails to capture the full variability present in the observations. This behavior is partly due to spatial averaging over model grid cells, which tends to smooth out extremes. However, counteracting this, so-called upscaling effects of the model tend to display small-scale physical behavior on the model scale. Thus, insufficient parameter perturbations may be another reason, together with the lack of subgrid-scale gravity waves and the use of climatologically prescribed aerosol fields, both of which constrain variability in ice nucleation conditions.

Moreover, the rank histogram reveals a slight negative bias, with observed  $RH_{ice}$  values more often exceeding the ensemble forecast range than falling below it. This suggests a systematic underestimation of  $RH_{ice}$  by the model, at least in parts of the  $RH_{ice} > 50\%$  regime. We found that this mainly occurs at ice supersaturated conditions. However, the rank histogram does not provide any information about magnitudes. Thus, we further analyze the ensemble's ability to classify ISSR and non-ISSR conditions below.

• Comment 9: Ligne 616: The mean diameter of ice crystals detrained from deep convection has been set to 200  $\mu$ m. Is this choice based on observations? If not, would it not be relevant to consider it as a tuning parameter by defining a plausible range of values?

**Response**

The choice of a mean ice crystal diameter of 200  $\mu$ m for detrained particles from deep convection is based on a combination of observational evidence and tuning experiments conducted during the development of the adapted two-moment microphysics scheme.

Observational studies suggest that ice crystal sizes in convective outflow typically range between 50 and 300  $\mu$ m (compare [7, 11, 10]). Within this plausible range, 200  $\mu$ m was found to yield consistent results in terms of cloud evolution and RHice behavior. Thus, this value provides a reasonable balance between physical realism and model performance.

• Comment 10: Lignes 629-630: The authors assume that the concentration of desert aerosols (Ndus) is constant at 200 hPa and set to 1000 m-3. Is this assumption realistic in all regions, particularly in tropical deep convection areas, where aerosol vertical distributions may vary due to convective transport processes? Wouldn't it be appropriate to set Ndus based on latitude and/or treat it as a parameter to be explored within a plausible range of values?

**Response**

We acknowledge that assuming a constant mineral dust concentration  $(N_{\rm dust})$  at 200 hPa may oversimplify regional and vertical variability, particularly in tropical deep convection areas where aerosol distributions are influenced by convective transport. While the Deutscher Wetterdienst (DWD) also operates ICON-ART, which treats mineral dust prognostically, our current setup uses a simplified approach for computational efficiency. Also importantly, not all dust particles act as ice-nucleating particles (INPs), especially in the tropics where many small particles may lack nucleating potential.

To account for uncertainty, we explored a wide range of  $N_{\rm dust}$  values (100–100.000 m-3) in tuning experiments and found that 1000 m-3 provides a reasonable balance between physical plausibility and model performance.

• Comment 11: Section 5.3: The authors discuss the model resolution and neighborhood considerations. In connection with Comment #2, could the authors comment on the uncertainties introduced in the results by the difference in resolution?

**Response**

Quantifying uncertainties is arguably a challenging endeavor. What we can say is that due to the lower resolution of the ICON model, localized patterns may occasionally be missed due to the averaging over a grid cell. In this regard, the model performance here represents a lower bound that might be improved with a finer grid.

**Manuscript changes**

In the "Discussion" section: We expect that using a finer grid for ICON predictions may enable such an approach, and most likely improve the overall verification scores.

**Specific Minor Comments**

1. Line 2. ISSR should be defined earlier as RHi>100%, preferably in the first sentence rather than in line 6

**Response**

We introduce ISSR and its definition now in the second sentence of the abstract

**Manuscript changes**

Contrails and contrail—induced cirrus clouds are considered the most significant non– $\rm CO_2$  contributors to aviation's climate impact. These clouds primarily form in ice–supersaturated regions (ISSRs), defined by relative humidity over ice (RHice) exceeding 100 %.

2. Line 23: Please specify the phenomena to which you are referring.

**Response**

We changed "These phenomena" to "These aircraft-induced clouds".

**Manuscript changes (Line 15)**

These aircraft–induced clouds present a complex challenge for climate assessment.

3. Line 59: The part of the sentence from "evaluated..." to "were found" is not very easy to read, as it presents metrics without explaining their implications and also uses acronyms without definitions. Since these details are not essential for understanding the rest of the paragraph,, I suggest that the authors replace this part with a more general formulation such as: "evaluated using accuracy assessment metrics."

**Response**

Thank you for the suggestion. We agree that the sentence could be made more accessible and have revised it to improve clarity while retaining the specific metrics used in the referenced study.

**Manuscript changes**

 $RH_{ice}$  predictions from IFS (Integrated Forecasting System), GFS (Global Forecast System), and S-WRF (a Weather Research and Forecasting model configuration by SATAVIA) were evaluated using standard classification metrics, including the  $F_1$  score and the Matthews Correlation Coefficient, which reflect the models' ability to correctly identify ice-supersaturated conditions. Moderate scores were found, indicating room for improvement in ISSR prediction skill.

4. In the caption of Fig. 1, please specify the altitude or pressure level corresponding to what you define as the near tropopause.

**Manuscript change**

- (a) Global forecast-only data of RHice near the tropopause (~10.2 km);
- 5. The results presented for the machine learning approach are very interesting. However, they could be better highlighted in a separate article where the method would be described in more detail, making it accessible and useful to a wider audience.

**Response**

We appreciate this suggestion and agree that a dedicated study focusing exclusively on the machine learning approach would be valuable and is indeed planned for future work. However, given the natural connection to the current study and the relevance of the results, we felt it was important to already present this first promising trial. Including it here demonstrates the potential of such methods in this context and lays the groundwork for more detailed investigations to follow.

**Manuscript changes**

We added a brief "Outlook" section, where we mention our initial machine learning results (previously part of the Discussion subsection) to emphasize their future potential.

**References**

- [1] Audran Borella, Étienne Vignon, Olivier Boucher, Yann Meurdesoif, and Laurent Fairhead. A new prognostic parameterization of subgrid ice supersaturation and cirrus clouds in the icolmdz agcm. *Journal of Advances in Modeling Earth Systems*, 17(8):e2024MS004918, 2025.
- [2] Audran Borella, Étienne Vignon, Olivier Boucher, and Susanne Rohs. An empirical parameterization of the subgrid-scale distribution of water vapor in the utls for atmospheric general circulation models. *Journal of Geophysical Research: Atmospheres*, 129(20):e2024JD040981, 2024.
- [3] Georgios Dekoutsidis, Silke Groß, Martin Wirth, Martina Krämer, and Christian Rolf. Characteristics of supersaturation in midlatitude cirrus clouds and their adjacent cloud-free air. *Atmospheric Chemistry and Physics*, 23(5):3103–3117, 2023.
- [4] Christoph Dyroff, Andreas Zahn, Emanuel Christner, Richard Forbes, Adrian M Tompkins, and Peter FJ van Velthoven. Comparison of ecmwf analysis and forecast humidity data with caribic upper troposphere and lower stratosphere observations. *Quarterly Journal of the Royal Meteorological Society*, 141(688):833–844, 2015.
- [5] K Gierens and P Spichtinger. On the size distribution of ice-supersaturated regions in the upper troposphere and lowermost stratosphere. In *Annales Geophysicae*, volume 18, pages 499–504. Springer, 2000.
- [6] Bernd Kärcher, EJ Jensen, GF Pokrifka, and Jerry Y Harrington. Ice supersaturation variability in cirrus clouds: Role of vertical wind speeds and deposition coefficients. *Journal of Geophysical Research: Atmospheres*, 128(22):e2023JD039324, 2023.
- [7] Martina Krämer, Christian Rolf, Anna Luebke, Armin Afchine, Nicole Spelten, Anja Costa, Jessica Meyer, Martin Zoeger, Jessica Smith, Robert L Herman, et al. A microphysics guide to cirrus clouds—part 1: Cirrus types. *Atmospheric Chemistry and Physics*, 16(5):3463–3483, 2016.
- [8] Konstantin Krüger, Andreas Schäfler, Martin Wirth, Martin Weissmann, and George C Craig. Vertical structure of the lower-stratospheric moist bias in the era5 reanalysis and its connection to mixing processes. *Atmospheric Chemistry and Physics*, 22(23):15559–15577, 2022.
- [9] Anne Kunz, Nicole Spelten, Paul Konopka, Rolf Müller, Richard M Forbes, and Heini Wernli. Comparison of fast in situ stratospheric hygrometer (fish) measurements of water vapor in the upper troposphere and lower stratosphere (utls) with ecmwf (re) analysis data. Atmospheric chemistry and physics, 14(19):10803–10822, 2014.
- [10] R Paul Lawson. Effects of ice particles shattering on the 2d-s probe. Atmospheric Measurement Techniques, 4(7):1361–1381, 2011.
- [11] R Paul Lawson, Eric Jensen, David L Mitchell, Brad Baker, Qixu Mo, and Bryan Pilson. Microphysical and radiative properties of tropical clouds investigated in tc4 and namma. *Journal of Geophysical Research: Atmospheres*, 115(D10), 2010.
- [12] Peter Spichtinger and Martin Leschner. Horizontal scales of ice-supersaturated regions. *Tellus B: Chemical and Physical Meteorology*, 68(1):29020, 2016.

- [13] Roger Teoh, Ulrich Schumann, Edward Gryspeerdt, Marc Shapiro, Jarlath Molloy, George Koudis, Christiane Voigt, and Marc EJ Stettler. Aviation contrail climate effects in the north atlantic from 2016 to 2021. Atmospheric Chemistry and Physics, 22(16):10919–10935, 2022.
- [14] Romy Ullrich, Corinna Hoose, Ottmar Möhler, Monika Niemand, Robert Wagner, Kristina Höhler, Naruki Hiranuma, Harald Saathoff, and Thomas Leisner. A new ice nucleation active site parameterization for desert dust and soot. *Journal of the Atmospheric Sciences*, 74(3):699–717, 2017.
- [15] Simon Unterstrasser and Klaus Gierens. Numerical simulations of contrail-to-cirrus transition—part 1: An extensive parametric study. *Atmospheric Chemistry and Physics*, 10(4):2017–2036, 2010.
- [16] WMO. Report of wmo's 2022 upper-air instrument intercomparison campaign. Instruments and Observing Methods Report 143, WMO, 2024.